# Depth is More Powerful than Width with Prediction Concatenation in Deep Forest

**Shen-Huan Lyu, Yi-Xiao He, Zhi-Hua Zhou**
National Key Laboratory for Novel Software Technology,
Nanjing University, Nanjing, 210023, China.
`{lvsh,heyx,zhouzh}@lamda.nju.edu.cn`

## Abstract

Random Forest (RF) is an ensemble learning algorithm proposed by Breiman [1] that constructs a large number of randomized decision trees individually and aggregates their predictions by naive averaging. Zhou and Feng [2] further propose Deep Forest (DF) algorithm with multi-layer feature transformation, which significantly outperforms random forest in various application fields. The prediction concatenation (PreConc) operation is crucial for the multi-layer feature transformation in deep forest, though little has been known about its theoretical property. In this paper, we analyze the influence of Preconc on the consistency of deep forest. Especially when the individual tree is inconsistent (as in practice, the individual tree is often set to be fully grown, i.e., there is only one sample at each leaf node), we find that the convergence rate of two-layer DF *w.r.t.* the number of trees $M$ can reach $\mathcal{O}(1/M^2)$ under some mild conditions, while the convergence rate of RF is $\mathcal{O}(1/M)$. Therefore, with the help of PreConc, DF with deeper layer will be more powerful than the shallower layer. Experiments confirm theoretical advantages.

## 1 Introduction

Random forest (RF) [1] is a state-of-art classification and regression algorithm that constructs a number of randomized individual decision trees during a parallel training phase and predicts by naive averaging the results. Recently, Zhou and Feng [2] propose Deep Forest (DF) algorithm with multi-layer feature transformation to investigate the possibility of tree-based representation learning. Benefiting from the multi-layer feature transformation in the cascade structure with prediction concatenation (PreConc), DF outperforms various tree-based algorithms [1, 3–5] in empirical study, and have been involved in real applications such as medicine [6], computer vision [7], remote sensing [8], financial fraud detection [9], etc. Numerous variants have been extended to various tasks and meet with remarkable success in [10–13]. There are also variants [14, 15] aiming at improving performance and reducing computational cost.

Empirical successes have attracted attention to the theoretical analysis of DF. Since its PreConc operation transforms features layer by layer, and each layer consists of complex non-parametric random forest estimators, analyzing the impact of the new features extracted from these estimators on generalization performance is important. To start theoretical analysis on DF, Lyu et al. [16] prove a margin-based generalization bound for additive new features in DF. Then, Pang et al. [14] prove an upper bound on the sample efficiency and inspire an efficient improvement for reducing computational cost of DF. In the direction of consistency, Arnould et al. [17] provide tight lower and upper bounds on the excess risk of a shallow centered random tree network, which leverages the new features to improve the performance of a single centered random tree. Assuming that the original and new features are separated and independently used in two stages, Lyu et al. [18] prove that the new features based on prediction are easy to cause overfitting risk, and propose to use the interaction of decision rules to alleviate it.

36th Conference on Neural Information Processing Systems (NeurIPS 2022).

However, previous theoretical studies often choose to ignore the impact of the new features on generating forests. For example, recent works [16, 18, 14] assume that the forest estimator in each layer is a black-box model and Arnould et al. [17] assume that there is only a single centered random tree in each layer. Such strong assumptions are in favor of theoretical analysis, but widen the gap between the theoretical results and practical deep forest architecture. To properly analyze DF, we must notice that, among the complex architecture in DF, both prediction concatenation (PreConc) [16] and the classification and regression trees (CART)-split criterion [19] play critical roles. Therefore, in order to open the black box of DF, we need to study the subtle combination of different techniques, i.e., analyze the properties of the new feature and study how CART generates and utilizes it.

**Contributions.** We compare the advantages of depth (cascade structure with PreConc in deep forest) over width (bagging of trees in random forest) in the scope of $\mathbb{L}_2^2$-consistency, based on the assumptions of the additive regression functions and uniform distribution over input space. The main contributions can be summarized as follows:

- We are the first to establish a consistency analysis of the prediction concatenation (PreConc) operation which is crucial for multi-layer feature transformation in deep forest, though based on a simplified version.

- We prove the universal consistency of deep forest when the total number of leaves of individual trees is chosen properly.

- In the practical setting, when the individual trees are fully grown, we prove that the convergence rate of two-layer deep forest reach $\mathcal{O}(1/M^2)$ *w.r.t.* the increasing number of trees $M$, while that of random forest is $\mathcal{O}(1/M)$. This result reflects that deep forest with deeper layer will be more powerful than shallower layer.

**Organization.** The rest of this work is organized as follows: Section 3 shows the setting and notations related to tree-based estimators in this work. Section 4 recalls the original deep forest architecture and simplifies it into a two-layer deep forest. Section 5 contains the properties of cascade structure with prediction concatenation. Section 6 proves the main result that depth is more powerful than width in deep forest architecture in the scope of consistency. Section 7 is devoted to the empirical studies of deep forest by verifying the theoretical results above. Section 8 concludes with future work. More experimental results and detailed proofs for theorems and propositions are given in the supplementary material due to page limitation.

## 2   Related work

Despite the widespread use and remarkable success of random forest in real world applications, the theoretical properties of it are still not fully understood [20, 21]. Breiman [1] offers an upper bound on the generalization error of random forest in terms of correlation and accuracy of the individual trees. Lin and Jeon [22] establish a connection between random forest and a particular class of nearest neighbor predictors, which are further studied by Biau and Devroye [23]. Meinshausen [24] studies the consistency of the quantile random forest for regression. In recent years, various theoretical works [20, 25–28] have been performed, analyzing the consistency of various simplified forests, and moving ever closer to practice. Denil et al. [21] narrows the gap between theory and practice of random forests for regression. Scornet et al. [29] prove the first $\mathbb{L}_2^2$-consistency of Breiman's original random forest with CART-split criterion based on the assumptions of the additive regression functions and uniform distribution over input space. Scornet [30] prove that infinite forest consistency implies finite forest consistency. Gao and Zhou [31] then present the convergence rate of purely randomized trees and a simplified variant of Breiman's original CART trees. Gao et al. [32] further expand it to multi-class setting. In addition, another research route is the theory analysis of feature importance [33–35]. Recently, Li et al. [36] derive non-asymptotic lower and upper bounds on the expected bias of MDI importance for random forests. Sutera et al. [37] establish a connection between MDI importance of pure random forest and Shapley values.

However, while deep forests further improve generalization performance, there is no theory to prove the advantages brought by depth. Therefore, studying the influence of depth is the theoretical cornerstone for distinguishing deep forests from random forests. For example, in the well-known deep neural networks (DNNs), there are a lot of theoretical works to study the effect of depth and width

on its representation ability and generalization performance, which show the theoretical advantages of deep neural networks over shallow neural networks [38–42]. These works all contribute to the understanding of deep learning and provide insight for designing algorithms.

## 3 Setting and notations

We first describe the setting and notations related to tree-based estimators in this work. For the sake of conciseness, we consider the regression setting.

**Setting.** We consider a regression framework, where the training set $S_n = \{(\boldsymbol{x}_1, y_1), \ldots, (\boldsymbol{x}_n, y_n)\}$ consists of $[0,1]^d \times \mathbb{R}$-valued independent random variables distributed as the prototype sample $(\boldsymbol{x}, y)$ with $\mathbb{E}[y^2] < \infty$. This underlying distribution, characterized by the marginal distribution $\mathcal{D}_{\mathcal{X}}$ on $[0,1]^d$ and by the conditional distribution $\mathcal{D}_{\mathcal{Y}|\mathcal{X}}$, can be written as

$$y = f(\boldsymbol{x}) + \epsilon \,, \tag{1}$$

where $f(\boldsymbol{x}) = \mathbb{E}[y|\boldsymbol{x}]$ is the conditional expectation of $y$ given $\boldsymbol{x}$, and $\epsilon$ is a noise satisfying $\mathbb{E}[\epsilon] = 0$ and $\mathrm{Var}[\epsilon] < \sigma^2$. The task considered in this paper is to output a randomized estimator $h_n(\cdot, \Theta, S_n)\colon [0,1]^d \to \mathbb{R}$ where $\Theta$ is a random variable that accounts for the randomization procedure and independent of the training set $S_n$. To simplify notation, we denote $h_n(\boldsymbol{x}, \Theta) = h_n(\boldsymbol{x}, \Theta, S_n)$. The quality of a randomized estimator $h_n$ is measured by its $\mathbb{L}_2$ risk

$$R(h_n) = \mathbb{E}\left[(h_n(\boldsymbol{x}, \Theta) - f(\boldsymbol{x}))^2\right] \,, \tag{2}$$

where the expectation is taken with respect to $(\boldsymbol{x}, \Theta)$, conditionally on $S_n$. As the training data size $n$ increases, we get a sequence of estimators $\{h_1, h_2, \ldots, h_n, \ldots\}$. A sequence of estimators $\{h_n\}_{n=1}^{\infty}$ is said to be consistent if $R(h_n) \to 0$ as $n \to \infty$.

**Trees and forests.** A random forest estimator $h_{M,n}(\boldsymbol{x}, \Theta)$ outputs the average prediction over $M$ individual randomized trees $h_n(\boldsymbol{x}, \Theta_j)$, $\forall j \in [M]$. Here, $[M] = \{1, 2, \ldots, M\}$ denotes the indexes of all individual randomized trees, where $\Theta_1, \ldots, \Theta_M$ are distributed identically and independently and denoted by a generic random variable $\Theta$. The random variable $\Theta$ can be used to sample the training set and select the candidate dimensions and positions for splitting. Specifically, a recursive partition $\Pi$ of $[0,1]^d$ is built by performing successive axis-aligned splits according to $\Theta$:

$$h_n(\boldsymbol{x}, \Theta) = \sum_{i=1}^{n} \frac{y_i \cdot \mathbb{1}(\boldsymbol{x}_i \in C_{\Pi,n}(\boldsymbol{x}))}{N_n(C_{\Pi,n}(\boldsymbol{x}))} \,, \tag{3}$$

where $C_{\Pi,n}(\boldsymbol{x})$ is the cell of the tree partition containing $\boldsymbol{x}$ and $N_n(C_{\Pi,n}(\boldsymbol{x}))$ is the number of training samples falling into $C_{\Pi,n}(\boldsymbol{x})$ with convention that the estimation equals to zero if the cell $C_{\Pi,n}(\boldsymbol{x})$ is empty. These trees are combined to form a finite forest estimation:

$$h_{M,n}(\boldsymbol{x}, \Theta) = h_{M,n}(\boldsymbol{x}, \Theta_1, \ldots, \Theta_M) = \frac{1}{M} \sum_{i=1}^{M} h_n(\boldsymbol{x}, \Theta_i) \,. \tag{4}$$

By the law of large numbers, for any fixed $\boldsymbol{x}$, conditionally on $\mathcal{D}_n$, the finite forest estimation converges to the infinite forest estimation:

$$h_{\infty,n}(\boldsymbol{x}) = \lim_{M \to \infty} h_{M,n}(\boldsymbol{x}, \Theta) = \mathbb{E}_{\Theta}[h_n(\boldsymbol{x}, \Theta)] \,. \tag{5}$$

When the number of samples tends to $\infty$, we denote by $h_{\infty}(\boldsymbol{x}, \Theta)$ the randomized tree with infinite samples and $h_{M,\infty}(\boldsymbol{x}, \Theta)$ the random forest with $M$ trees and infinite samples.

## 4 Deep forest

We recall the original deep forest algorithm in Section 4.1, and describe the simplified two-layer deep forest algorithm in Section 4.2.

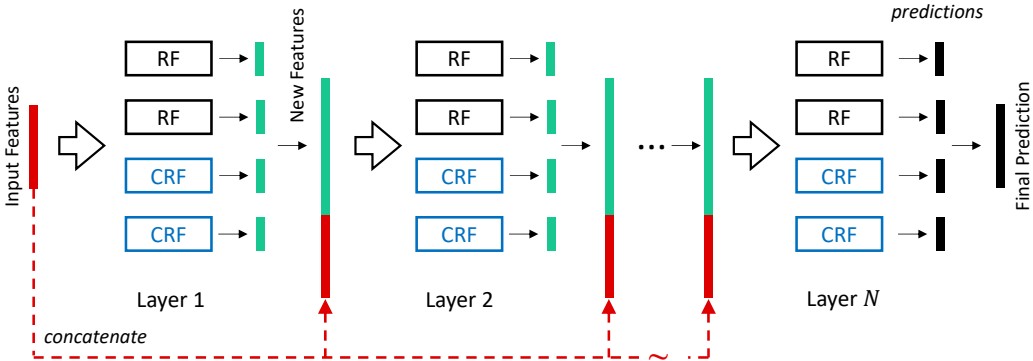

Figure 1: Deep forest architecture (the scheme is taken from Zhou and Feng [2]).

## 4.1 Original deep forest

Deep forest [2] is a tree-based deep model made up of non-differentiable forest modules without back-propagation. Each layer of DF is composed of two Breiman's random forest (RF) and two Completely-Random Forest (CRF). The RF is composed of CARTs and CRF is composed of pure random trees. In the regression setting, each forest of each layer outputs a prediction for any query point $\boldsymbol{x}$, corresponding to the average response in the leaf node (cell) containing $\boldsymbol{x}$. At a given layer, the predictions of all forests of this layer are concatenated together with raw features. This prediction concatenation process is called PreConc, which is repeated for each layer until the best layer and construct a deep forest. For an overview of the architecture of deep forest, we refer readers to the work [43, 2].

## 4.2 Simplified two-layer deep forest

In order to narrow the gap between theoretical and empirical studies of deep forest, we retain CART-split criterion and PreConc when simplifying the model. We define a simplified two-layer deep forest, whose each layer is composed of one Breiman's random forest. We denote by $h_{M,n}^{(1)}(\boldsymbol{x}, \Theta, S_n)$ the first-layer forest estimator and $h_{M,n}^{(2)}([\boldsymbol{x}, h], \Theta, S_n)$ the second-layer forest estimator, where $[\boldsymbol{x}, h]$ is the concatenation of the raw features $\boldsymbol{x}$ and the new feature $h$. Then we can define the two-layer deep forest as follows:

$$\bar{h}_{2M,n}(\boldsymbol{x}, \Theta, S_n) = h_{M,n}^{(2)} \circ h_{M,n}^{(1)}(\boldsymbol{x}, \Theta, S_n) . \tag{6}$$

The complete algorithm is shown in Algorithm 1. This algorithm has three parameters: $m_{\text{try}} \in \{1, \ldots, d\}$ is the number of pre-selected directions for splitting, $a_n \in \{1, \ldots, n\}$ is the number of samples in each tree, $t_n \in \{1, \ldots, a_n\}$ is the number of leaves in each tree. In the default procedure, the parameters are set as follows: $m_{\text{try}}$ is set to $d$. $a_n$ is set to $n/k$, where $k$ is the $k$-fold cross-validation in Zhou and Feng [2]'s deep forest. $t_n = a_n$ means we use fully grown CART. Notice that $k$ controls the subsampling rate $a_n/n = 1/k$, which is proved by Scornet et al. [29] to be the key component for imposing tree diversity.

Given the above parameters, the most basic part of Algorithm 1 is the training process of each CART. Let $C$ be a cell and $N_n(C)$ be the number of data points falling in $C$. A split in $C$ is a pair $(j, z)$, where $j$ is a dimension in $\{1, \ldots, d\}$ and $z$ is the position of the split along the $j$-th dimension in cell $C$. Let $\mathcal{S}_C$ be the set of all possible splits in $C$. The CART-split criterion [19] takes the form

$$\hat{L}_n(j, z) = \frac{1}{N_n(C)} \sum_{i: \boldsymbol{x}_i \in C} (y_i - \mu_n(C))^2$$
$$- \frac{1}{N_n(C_L)} \sum_{i: \boldsymbol{x}_i \in C_L} (y_i - \mu_n(C_L))^2 - \frac{1}{N_n(C_R)} \sum_{i: \boldsymbol{x}_i \in C_R} (y_i - \mu_n(C_R))^2 , \tag{7}$$

where $C_L = \{\boldsymbol{x} \in C : \boldsymbol{x}^{(j)} < z\}$, $C_R = \{\boldsymbol{x} \in C : \boldsymbol{x}^{(j)} \geq z\}$, and $\mu_n(C) = \frac{1}{N_n(C)} \sum_{i: \boldsymbol{x} \in A} y_i$ denotes the average response in any cell $C$ (resp. $\mu_n(C_L)$, $\mu_n(C_R)$), with the convention $0/0 = 0$.

**Algorithm 1:** A simplified variant of Zhou's original deep forest

---

**Require:** A training set $S_n = \{(\boldsymbol{x}_1, y_1), \ldots, (\boldsymbol{x}_n, y_n)\}$, the number of trees $2M$,
  $m_{\text{try}} \in \{1, \ldots, d\}$, $a_n \in \{1, \ldots, n\}$, $t_n \in \{1, \ldots, a_n\}$ and the query point $\boldsymbol{x} \in [0,1]^d$.
**Ensure:** Two-layer deep forest $\bar{h}_{2M,n}(\cdot) = h_{M,n}^{(2)} \circ h_{M,n}^{(1)}(\cdot)$ and the prediction of $\boldsymbol{x}$.

1: **for** layer $\ell$ in $\{1, 2\}$ **do**
2:   **if** $\ell = 2$ **then**
3:     $S_n \leftarrow \{([\boldsymbol{x}_1, h_{M,n}^{(1)}(\boldsymbol{x}_1)], y_1), \ldots, ([\boldsymbol{x}_n, h_{M,n}^{(1)}(\boldsymbol{x}_n)], y_n)\}$.     ▷ *Prediction concatenation.*
4:   **end if**
5:   **for** tree $j \in \{1, 2, \ldots, M\}$ **do**
6:     Select $a_n$ data points, without replacement, uniformly in $S_n$.
7:     Set $\Pi_0 = \{[0,1]^d\}$ the partition associated with the root of the tree.
8:     For all $1 \leq i \leq a_n$, set $\Pi_i = \emptyset$.
9:     Set $n_{\text{nodes}} = 1$ and level $= 0$.
10:    **while** $n_{\text{nodes}} < t_n$ **do**
11:      **if** $\Pi_{\text{level}} = \emptyset$ **then**
12:        level $=$ level $+ 1$.
13:      **else**
14:        Let $C$ be the first element in $\Pi_{\text{level}}$.
15:        **if** $C$ contains exactly one point **then**
16:          $\Pi_{\text{level}} \leftarrow \Pi_{\text{level}} \setminus \{C\}$.
17:          $\Pi_{\text{level}+1} \leftarrow \Pi_{\text{level}+1} \cup \{C\}$.
18:        **else**
19:          Select the best split $(j_n^*, z_n^*)$ in $C$ by optimizing the CART-split criterion along the
             dimension $D$ in $\{1, \ldots, d\}$.                    ▷ *See details in Eq.* (7).
20:          Split cell $C$ along $D$ according to the best split $(j_n^*, z_n^*)$. Call $C_L$ and $C_R$.
21:          $\Pi_{\text{level}} \leftarrow \Pi_{\text{level}} \setminus \{C\}$.
22:          $\Pi_{\text{level}+1} \leftarrow \Pi_{\text{level}+1} \cup \{C_L\} \cup \{C_R\}$.
23:          $n_{\text{nodes}} \leftarrow n_{\text{nodes}} + 1$.
24:        **end if**
25:      **end if**
26:    **end while**
27:    Compute the predicted value $h_n^{(\ell)}(\boldsymbol{x}, \Theta_j, S_n)$ at the query point $\boldsymbol{x}$ equaling the average of
       the $Y_i$'s falling in the cell of $\boldsymbol{x}$ in partition $\Pi_{\text{level}} \cup \Pi_{\text{level}+1}$.
28:  **end for**
29:  Compute the random forest estimation $h_{M,n}^{(\ell)}(\boldsymbol{x}, \Theta, S_n)$ at the query point $\boldsymbol{x}$ according to
     Eq. (4).
30: **end for**
31: Compute the two-layer deep forest estimation $\bar{h}_{2M,n}(\boldsymbol{x}, \Theta, S_n)$ at the query point $\boldsymbol{x}$ according
    to Eq. (6).

---

At each cell $C$, the best split $(j_n^*, z_n^*)$ is selected by maximizing $\hat{L}_n(j, z)$ over $\mathcal{M}_{\text{try}}$ and $\mathcal{S}_C$, that is,

$$(j_n^*, z_n^*) \in \underset{\substack{(j^*, z^*) \in \mathcal{S}_C \\ j \in \mathcal{M}_{try}}}{\arg \max} \ \hat{L}_n(j, z) \,. \tag{8}$$

## 5 Properties of prediction concatenation

In this section we show that the properties of the simplified deep forest enable us to analyze the influence of the concatenated new feature in deep forest and the local variation related to the empirical CART-split criterion.

We consider an additive regression model satisfying the following assumption:

**Assumption 1.** *The response $y$ follows*

$$y = \sum_{j=1}^{d} f_j(x^{(j)}) + \epsilon \,, \tag{9}$$

*where $\boldsymbol{x} = \left(x^{(1)}, \ldots, x^{(d)}\right)$ is uniformly distributed over $[0, 1]^d$, $\epsilon$ is an independent centered Gaussian noise with finite variance $\sigma^2 > 0$ and each component $f_j$ is continuous.*

Stone [44], Hastie and Tibshirani [45] popularize these models, which decompose the regression function as a sum of univariate functions. Especially, Scornet et al. [29] prove the consistency of Breiman's original random forest under this assumption. On this basis, we study the impact of the new feature generated in the deep forest.

To start with, we analyze the priority of the new features generated by the previous layer in the selection of splitting features under the CART-split criterion, in both infinite sample regime and finite sample regime respectively.

**Proposition 1** (Priority of the new feature). *Assume the data set follows Assumption 1 and the first-layer forest is consistent. The following results hold for any CART in the second-layer forest.*

1. *In the infinite sample regime ($n = \infty$), we consider a single second-layer CART $h_\infty^{(2)}(\cdot, \Theta)$. All splits in this CART are performed along the new feature only.*

2. *In the finite sample regime ($n < \infty$), we consider a single second-layer CART $h_n^{(2)}(\cdot, \Theta)$. Fix $k \in \mathbb{N}^*$ and $\xi, \rho > 0$. Then, with probability $\geq 1 - \rho$, for all $n$ large enough, we have, the error of the first-layer forest is bound by $\xi$. As a consequence, the first $k$ splits $(j_{q,n}(\boldsymbol{x}), 1 \leq q \leq k)$ in this CART are performed along the new feature only.*

*Proof sketch.* **(P1.1).** In the infinite sample regime, the random forest estimation of the first layer has zero error. Therefore, the new feature of the second layer is the target function $h_{M,\infty}^{(1)}(\boldsymbol{x}) = f(\boldsymbol{x})$. Obviously, the new feature is the most informative dimension. Therefore, when splitting in any cell, CART algorithm will select the new feature as the splitting dimension. **(P1.2).** In the finite sample regime, the random forest estimation of the first layer is not precise. Therefore, there is an error related to $n$ between the new feature and the target function. Firstly, we prove that the distance between the theoretical ($n = \infty$) and empirical ($n < \infty$) first $k$ splits of the CART algorithm is bounded by $c\xi$ with probability $\geq 1 - \rho$, when $n$ is large enough. Connecting with the result of theoretical split above, the proof is completed. □

**Remark 1.** Proposition 1 shows that the trees in the second layer primarily choose the new feature to split and the degree of this priority depends on the error of the first-layer forest estimator. This also reveals that the advantages of deep forest depend on the performance of the first layer. If the forest of the first layer does not return an estimation with noise reduction, the performance of deep forest cannot be further improved through PreConc operation. This is consistent with the empirical results in previous work [2, 17].

In order to control the risk of deep forest, we need study the local variation property of the empirical CART-split criterion. For any cell $C$, the variation of regression function $f(\boldsymbol{x})$ within $C$ is defined as

$$\Delta(f, C) = \sup_{\boldsymbol{x}, \boldsymbol{x}' \in C} |f(\boldsymbol{x}) - f(\boldsymbol{x}')| . \tag{10}$$

**Proposition 2** (Variation of $f$ in the empirical cell). *Assume that Assumption 1 holds and the first-layer forest is consistent. The following results hold for any CART in the second-layer forest $h_n^{(2)}(\cdot, \Theta)$. After splitting along the new feature, the CART will estimate the residual of the first-layer forest estimation. Then for all $\rho, \xi > 0$, there exists $N \in \mathbb{N}^*$ such that, for all $n > N$,*

$$\Pr\left[\Delta\left(f, C_{\Pi,n}(\boldsymbol{x}, \Theta)\right) \leq \xi\right] \geq 1 - \rho . \tag{11}$$

*Proof sketch.* Firstly, we prove the variation of $f(\boldsymbol{x})$ within the cell obtained by the theoretical CART-split criterion converges to zero. Secondly, we prove that the distance between the theoretical and empirical first $k$ splits of the CART convergence to zero. Finally, we prove that the variation of $f(\boldsymbol{x})$ within the empirical cell is close to the theoretical cell. □

Proposition 2 shows that the variation of the regression function $f(\boldsymbol{x})$ within a cell of a random tree $C_{\Pi,n}$ is small provided $n$ is large enough, thereby forcing the approximation error of DF to asymptotically approach zero.

# 6 Depth is more powerful than width

Our first result considers the $t_n < a_n$ regime, where the number of samples in each leaf node tends to $\infty$ as $t_n \to \infty$ and $a_n \to \infty$. We prove that controlling the depth of the trees through the number of leaves $t_n$ is sufficient to achieve consistency of deep forest.

**Theorem 3** (Universal consistency). *Let $M \geq 1$. Consider two-layer deep forest $\bar{h}_{2M,n}$ given by Eq. (6) and Breiman's random forest $h_{M,n}$ given by Eq. (4) for the random CARTs satisfying $a_n \to \infty, t_n \to \infty$ and $t_n(\log a_n)^9/a_n \to 0$. Then under the setting described in Section 3 and assume the data set follows Assumption 1,*

1. *[29, Theorem 1] the Breiman's random forest $h_{2M,n}$ is consistent for any $M \geq 1$,*

2. *the two-layer deep forest $\bar{h}_{M,n}$ is consistent for any $M \geq 1$.*

*Proof sketch.* **(T3.1).** The universal consistency of Breiman's random forest is proved by Scornet et al. [29]. **(T3.2).** Firstly, we already know that the variation of the target regression function $f(\boldsymbol{x})$ within a cell of a randomized CART in the second-layer forest is small when $n$ is large enough via Proposition 2. Similar as Scornet et al. [29], we utilize Proposition 2 to control the approximation error of the two-layer deep forest. And the parameter $t_n$ allows us to control the size of the leaves of CART, which allows us to have enough samples in each leaf node to smooth the impact of noise, so as to control the estimation error. Connecting the approximation and estimation error, the consistency of a CART of second-layer deep forest is proved. The universal consistency can be proved via [25, Proposition 1], which guarantees that the error of forest estimator is no more than twice that of individual randomized CART. □

**Remark 2.** Notice that under this setting, random forest and even deep forest have no obvious advantages over single CART in theory. When we use forests in practice, we do not choose to control the depth of the trees. Empirical studies in Section 7.2 show that the forest with $t_n = a_n$ always outperforms the forest with $t_n < a_n$. Actually, the fully grown trees $t_n = a_n$ is the setting close to practical forest algorithm.

In order to deal with the $t_n = a_n$ regime, we need to introduce an assumption first proposed by Scornet et al. [29]. We denote by $Z_i = \mathbb{1}(\boldsymbol{x}_i \in C_{\Pi,n}(\boldsymbol{x}))$ the indicator that $\boldsymbol{x}_i$ falls into the same cell as $\boldsymbol{x}$ in the random tree designed with $\mathcal{D}_n$ and the random parameter $\Theta$. $Z'_j = \mathbb{1}(\boldsymbol{x}_i \in C_{\Pi'}(\boldsymbol{x}))$ is another independent indicator. Then, we define the correlation between these two indicators conditionally on $y_i, y_j$ or not, respectively

$$\phi_{i,j}(y_i, y_j) = \mathbb{E}[Z_i, Z'_j | \boldsymbol{x}, \boldsymbol{x}_1, \ldots, \boldsymbol{x}_n, y_i, y_j] \quad \text{and} \quad \phi_{i,j} = \mathbb{E}[Z_i, Z'_j | \boldsymbol{x}, \boldsymbol{x}_1, \ldots, \boldsymbol{x}_n] . \quad (12)$$

**Assumption 2.** *Let $Z_{i,j} = (Z_i, Z_j)$. Then one of the following two conditions holds:*

1. *One has*

$$\lim_{n\to\infty} (\log a_n)^{2d-2}(\log n)^2 \mathbb{E}\left[\max_{i \neq j} |\phi_{i,j}(y_i, y_j) - \phi_{i,j}|\right]^2 = 0 . \quad (13)$$

2. *There exists a constant $C > 0$ and a sequence $(\gamma_n)_n \to 0$ such that, almost surely,*

$$\max_{\ell_1,\ell_2=0,1} \frac{|\mathrm{Cor}\left[(y_i - f(\boldsymbol{x}_i)), \mathbb{1}(Z_{i,j} = (\ell_1, \ell_2)|\boldsymbol{x}_i, \boldsymbol{x}_j, y_j)]\right|}{\mathrm{Pr}^{1/2}[Z_{i,j} = (\ell_1, \ell_2)|\boldsymbol{x}_i, \boldsymbol{x}_j, y_j]} \leq \gamma_n , \quad (14)$$

*and*

$$\max_{\ell_1=0,1} \frac{|\mathrm{Cor}\left[(y_i - f(\boldsymbol{x}_i))^2, \mathbb{1}(Z_i = \ell_1|\boldsymbol{x}_i)]\right|}{\mathrm{Pr}^{1/2}[Z_i = \ell_1|\boldsymbol{x}_i]} \leq C . \quad (15)$$

**(A2.1.)** means that the influence of two labels $y_i, y_j$ on the probability of connection of two couples of random points converge to zero as $n \to \infty$. **(A2.2.)** means that the correlation between the noise and the probability of connection of two couples of random points vanishes quickly enough, as $n \to \infty$. However, this assumption is too strong for the Breiman's original random forest [1]. Scornet et al. [29] emphasize that they cannot know whether or not Assumption 2 is satisfied in random forest. In this paper, we recall this assumption and state that this assumption is mild for the second-layer forest estimation in deep forest algorithm.

Since deep forest concatenates the prediction with the raw features as the input for the next layer, the label information is encoded into the new feature of second layer. In this way, the influence of two labels on the connection probability of this pair of samples tends to zero, so **(A2.1.)** is mild for the second-layer forest estimation. From another point of view, Proposition 1, the priority of new feature also shows that the partition can be independent of labels, and the information of new feature is enough to obtain appropriate partition results, so **(A2.2.)** is mild too.

**Theorem 4** (Depth is more powerful than width). *Let $M \geq 1$. Consider two-layer deep forest $\bar{h}_{2M,n}$ given by Eq. (6) and Breiman's random forest $h_{M,n}$ given by Eq. (4) for the random CARTs satisfying $a_n \to \infty, t_n \to \infty$, $t_n = a_n$ and $a_n \log n / n \to 0$. Then under the setting described in Section 3 and assume the data set follows Assumption 1 and 2, the following results hold*

1. *[29, Theorem 2] [30, Theorem 3] The Breiman's random forest $h_{\infty,n}$ is consistent, and for all $M, n \in \mathbb{N}$,*

$$0 \leq R(h_{2M,n}) - R(h_{\infty,n}) \leq \frac{8\|f\|_\infty^2 + 8\sigma^2(1 + 4\log n)}{M} . \tag{16}$$

2. *The two-layer deep forest $\bar{h}_{\infty,n}$ is consistent, and for all $M, n \in \mathbb{N}$, if $\Delta(f, C_{\Pi,n}(\boldsymbol{x}, \Theta))$ is small enough, then*

$$0 \leq R(\bar{h}_{2M,n}) - R(\bar{h}_{\infty,n}) \leq \frac{64\|f\|_\infty^2 + 64\sigma^2(1 + 4\log n)}{M^2} . \tag{17}$$

*Proof sketch.* **(T4.1).** The consistency of the infinite Breiman's random forest is proved by Scornet et al. [29]. And the convergence rate of the finite random forest with the number of trees $M$ is proved by Scornet [30]. **(T4.2).** Similar to Scornet et al. [29], we recall Proposition 2 to control the approximation error of the two-layer deep forest. Then the estimation error is controlled by forcing the subsampling rate $a_n/n$ to be $o(1/\log n)$. Different from the bagging-style mechanism in random forest, the residual-style mechanism shown in Proposition 2 makes the second-layer forest in DF can reuse the first-layer estimation and focus on the residual learning. □

**Remark 3.** In the $t_n = a_n$ setting, Scornet et al. [29] show that the sub-sampling rate $a_n/n$ is the key component in random forest. Because the small rate ensures that query point $\boldsymbol{x}$ is connected with enough different data points through different trees, the convergence rate of RF is $\mathcal{O}(1/M)$ w.r.t. the number of trees $M$. Theorem 4 proves that, if the first layer forest can encode the regression function $f(\boldsymbol{x})$ into the new feature with noise reduction, the cascade structure with PreConc in DF can further accelerate the convergence. Because the second layer forest estimates the residual of the first layer, the trees in each layer of forest are more different. As a result, the convergence rate of deep forest will be improved to $\mathcal{O}(1/M^2)$. This result reflects that deep forest with deeper layer will be more powerful than shallower layer.

## 7 Simulation experiments

### 7.1 Priority of the new feature

This experiment aims to verify the priority of the new feature in choosing which feature to split as suggested in Proposition 1. We focus on a second layer decision tree built upon the first layer random forest. Since a regression forest has only one output dimension, there is only one new feature for the second layer tree. More specifically, we count the maximum consecutive levels from root node that use the new feature to split, which we call *effective depth* for short.

The synthetic data set is generated as $y = f(\boldsymbol{x}) + \epsilon$, where

$$f(\boldsymbol{x}) = \frac{1}{5} \sum_{1 \leq j \leq 5} x_j , \tag{18}$$

$\boldsymbol{x}$ is uniformly distributed over $[0,1]^5$, $\epsilon \sim \mathcal{N}\left(0, \sigma^2\right)$, where $\sigma = 0.02$. We vary the number of training samples and the number of trees in the first layer forest, and report the average effective depth of 5 runs in Figure 2(a). It is easy to observe that no matter what the first layer's setting is, the effective depth of new feature is at least 2.4. That is to say, at the beginning of the growing of the second layer tree, CART will always choose the new feature to split. And we can see that with

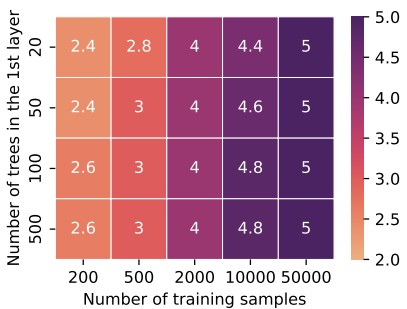

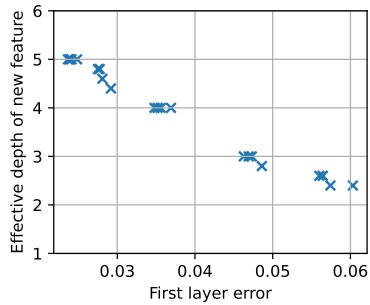

(a) Heatmap of the effective depth of new feature under different settings of the first layer forest.

(b) Effective depth of new feature against the predictive error of the first layer.

Figure 2: Illustrations of the effective depth of the new feature (the consecutive levels from root node that split on the new feature only). The larger the effective depth, the higher priority the new feature takes in being chosen as the split feature under the CART-split criterion.

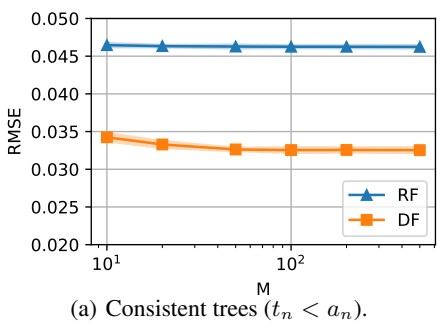

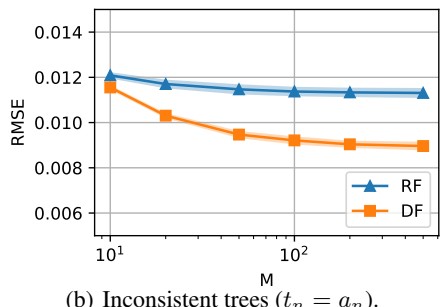

(a) Consistent trees ($t_n < a_n$).

(b) Inconsistent trees ($t_n = a_n$).

Figure 3: Root mean square error with the increasing of trees.

## 7.2 Convergence rate *w.r.t.* the number of trees

To study the convergence rate of predictive error with respect to the number of trees in the whole model, we fix the number of training samples to be 50,000. We compare a 2-layer deep forest (DF) with $M$ trees each layer to a random forest (RF) with $2M$ trees, and we adopt 3-fold split in training both DF and RF to ensure fairness. The performance is measured by RMSE with respect to the noise-free version of data generating function. Since the training data are with noise, a fully grown tree is inconsistent. In Figure 3(a), we set the minimum leaf size of the trees to be $\sqrt{n}$, i.e., 233 in the case when training data size is 50,000. In Figure 3(b) the trees are fully grown with only one sample in each leaf. We plot the average RMSE of 5 runs against the increasing number of trees, with the colored band indicating the standard deviation.

Theorem 3 reveals that when the component trees are consistent, random forest and deep forest are both consistent. However, the consistency analysis result cannot guarantee the finite sample performance in practice. Comparing Figure 3(a) to Figure 3(b), we can see that even though the training set is as large as 50,000, the performances of RF and DF using consistent trees are still much worse than using inconsistent trees. This observation confirms the efficacy of the default experimental setting that uses fully grown trees in RF and DF. Figure 3(b) shows that DF enjoys a faster improvement in RMSE with the increasing of $M$. More specifically, DF with $M = 20$ outperforms RF with $M = 500$. This experimental result matches our theoretical result in Theorem 4 that DF has a faster convergence rate *w.r.t.* the number of trees $M$.

# 8 Conclusion

In this paper, we prove that a two-layer deep forest has a faster convergence rate *w.r.t.* the number of component trees $M$ than random forest. This work provides the first theoretical analysis of the prediction concatenation (PreConc) operation which is crucial for feature transformation in deep forests, although based on a very simplified structure where the concatenation of multiple random forests' predictions and completely-random forests' predictions in each layer of deep forest has not been taken into account.

On the one hand, this paper focuses on the asymptotic consistency of deep forests, so the result is strictly true only when the number of samples tends to infinity. As for the generalization analysis of deep forests with finite samples, we leave it to future work. On the other hand, the two assumptions used in this paper have certain limitations. Experiments on simulation and real-world data sets show that our theoretical results are valid in many objective function classes other than Assumption 1. How to further relax the conditions in Assumption 1 will be an interesting problem. As for Assumption 2, it is still not strictly verified. However, quantifying the trade-off between label information and partition randomness will be a very important topic in future work.

## Acknowledgements

This research was supported by the NSFC (61921006) and the Collaborative Innovation Center of Novel Software Technology and Industrialization. The authors would like to thank the anonymous reviewers for constructive suggestions, as well as Jin-Hui Wu, Qin-Cheng Zheng and Peng Tan for helpful discussions.

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
