the increase of trees in the first layer and training samples, the CART's preference of new feature extend to deeper layers. Figure 2(b) further plots the effective depth against the predictive error of the first layer measured by root mean square error (RMSE), showing that the better the prediction performance of the first layer, the more favored the new feature is for splitting.

### 7.2 Convergence rate *w.r.t.* the number of trees

To study the convergence rate of predictive error with respect to the number of trees in the whole model, we fix the number of training samples to be 50,000. We compare a 2-layer deep forest (DF) with $M$ trees each layer to a random forest (RF) with $2M$ trees, and we adopt 3-fold split in training both DF and RF to ensure fairness. The performance is measured by RMSE with respect to the noise-free version of data generating function. Since the training data are with noise, a fully grown tree is inconsistent. In Figure 3(a), we set the minimum leaf size of the trees to be $\sqrt{n}$, i.e., 233 in the case when training data size is 50,000. In Figure 3(b) the trees are fully grown with only one sample in each leaf. We plot the average RMSE of 5 runs against the increasing number of trees, with the colored band indicating the standard deviation.

Theorem 3 reveals that when the component trees are consistent, random forest and deep forest are both consistent. However, the consistency analysis result cannot guarantee the finite sample performance in practice. Comparing Figure 3(a) to Figure 3(b), we can see that even though the training set is as large as 50,000, the performances of RF and DF using consistent trees are still much worse than using inconsistent trees. This observation confirms the efficacy of the default experimental setting that uses fully grown trees in RF and DF. Figure 3(b) shows that DF enjoys a faster improvement in RMSE with the increasing of $M$. More specifically, DF with $M = 20$ outperforms RF with $M = 500$. This experimental result matches our theoretical result in Theorem 4 that DF has a faster convergence rate *w.r.t.* the number of trees $M$.

# 8 Conclusion

In this paper, we prove that a two-layer deep forest has a faster convergence rate *w.r.t.* the number of component trees $M$ than random forest. This work provides the first theoretical analysis of the prediction concatenation (PreConc) operation which is crucial for feature transformation in deep forests, although based on a very simplified structure where the concatenation of multiple random forests' predictions and completely-random forests' predictions in each layer of deep forest has not been taken into account.

On the one hand, this paper focuses on the asymptotic consistency of deep forests, so the result is strictly true only when the number of samples tends to infinity. As for the generalization analysis of deep forests with finite samples, we leave it to future work. On the other hand, the two assumptions used in this paper have certain limitations. Experiments on simulation and real-world data sets show that our theoretical results are valid in many objective function classes other than Assumption 1. How to further relax the conditions in Assumption 1 will be an interesting problem. As for Assumption 2, it is still not strictly verified. However, quantifying the trade-off between label information and partition randomness will be a very important topic in future work.

## Acknowledgements

This research was supported by the NSFC (61921006) and the Collaborative Innovation Center of Novel Software Technology and Industrialization. The authors would like to thank the anonymous reviewers for constructive suggestions, as well as Jin-Hui Wu, Qin-Cheng Zheng and Peng Tan for helpful discussions.

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

# Supplementary Materials for
# Depth is More Powerful than Width in Deep Forest

## A    Table of notations

| Category | Sign | Description |
|---|---|---|
| Setting | $\mathcal{D}$ | The unknown distribution |
| | $f(\boldsymbol{x})$ | The target regression function |
| | $\epsilon$ | The Gaussian noise |
| | $S_n$ | Data set with $n$ samples |
| | $(\boldsymbol{x}, y)$ | A sample drawn from $\mathcal{D}$ |
| | $R(h)$ | $\mathbb{L}_2$ risk of the estimator $h$ |
| Tree and forest | $\Theta$ | A random variable that accounts for the randomization procedure and independent of $S_n$ |
| | $h_n(\boldsymbol{x}, \Theta, S_n)$ | A randomized decision tree |
| | $\Pi$ | The partition built by performing successive axis-aligned splits according to $\Theta$ |
| | $C_{\Pi,n}(\boldsymbol{x})$ | The cell of the tree partition containing $\boldsymbol{x}$ |
| | $N_n(C_{\Pi,n}(\boldsymbol{x}))$ | The number of training samples falling into $C_{\Pi,n}(\boldsymbol{x})$ |
| | $h_{M,n}(\boldsymbol{x}, \Theta, S_n)$ | A random forest |
| | $W_{ni}(\boldsymbol{x})$ | The probability that $\boldsymbol{x}$ and $\boldsymbol{x}_i$ fall into a same cell. |
| | $\mathbb{1}_{\boldsymbol{x} \overset{\Theta}{\leftrightarrow} \boldsymbol{x}_i}$ | The indicator where $\boldsymbol{x}$ and $\boldsymbol{x}_i$ are connected. |
| Deep forest | $[\boldsymbol{x}, h]$ | PreConc: The concatenation of the raw features $\boldsymbol{x}$ and the new feature (prediction) $h$ |
| | $\bar{h}_{2M,n}(\boldsymbol{x}, \Theta, S_n)$ | A two-layer deep forest with $2M$ trees |
| | $h_{M,n}^{(\ell)}(\boldsymbol{x}, \Theta, S_n)$ | The $\ell$-th layer forest of two-layer deep forest with $M$ trees, $\ell \in \{1, 2\}$ |
| CART-split | $t = (j, z)$ | A split where $j$ is a dimension in $\{1, \ldots, d\}$ and $z$ is the position of the split along the $j$-th dimension |
| | $C, C_L$ and $C_R$ | Any cell $C$, its left node and right node |
| | $\mu_n(\cdot)$ | The average response in any cell |
| | $\hat{L}_n(j, z)$ | The empirical version of CART-split criterion function |
| | $(j_n^*, z_n^*)$ | The best split selected by maximizing $\hat{L}_n(j, z)$ |
| | $L^*(j, z)$ | The theoretical version of CART-split criterion function |
| | $(j^*, z^*)$ | The best split selected by maximizing $L_n^*(j, z)$ |
| | $\boldsymbol{t}_k = (t_1, \ldots, t_k)$ | A sequence of first $k$ splits |
| | $\mathcal{T}_k(\boldsymbol{x})$ | The collection of all possible $k \geq 1$ consecutive splits used to build the cell containing $\boldsymbol{x}$ |
| | $d_\infty(\boldsymbol{t}_k, \mathcal{T})$ | The infinite distance $d_\infty$ between $\boldsymbol{t}_k \in \mathcal{T}_k(\boldsymbol{x})$ and any $\mathcal{T} \subset \mathcal{T}_k(\boldsymbol{x})$ |
| Statistics | $\mathrm{Cor}(x_1, x_2)$ | The correlation between $x_1$ and $x_2$ |
| | $\mathrm{Var}(x)$ | The variance of $x$ |
| | $\mathrm{Unif}([0, 1]^d)$ | The uniform distribution on $[0, 1]^d$ |

Table S1: Notations of this work.

# B   Proofs

In this section, we provide the detailed proofs for the main theorems and corollaries. First, we present a series of useful lemmas as follows:

**Lemma 5.** *For any $a, b, z \in \mathbb{R}$, let $f(x)$ be a continuous and bounded function, and $F(z) = \int_a^z f(t) \, dt$ be the integral function of $f(x)$ over $a$ to $z$, and given $\int_a^b f(t) \, dt = 0$. We denote by $\mu_{a \leq x \leq z}$ the average of $f(x)$ between $a$ and $z$, and $\mu_{z \leq x \leq b}$ the average of $f(x)$ between $z$ and $b$.*

$$\frac{1}{(z-a)(b-z)} F^2(z) = \frac{(z-a)(b-z)}{(b-a)^2} (\mu_{a \leq x \leq z} - \mu_{z \leq x \leq b})^2 . \tag{19}$$

**Proof of Lemma 5.**

$$
\begin{aligned}
\frac{1}{(z-a)(b-z)} F^2(z) =& \frac{z-a}{b-a} (F(z)/(z-a))^2 + \frac{b-z}{b-a} (F(z)/(b-z))^2 \\
=& \left( \frac{z-a}{b-a} \mu_{a \leq x \leq z}^2 + \frac{b-z}{b-a} \mu_{z \leq x \leq b}^2 \right) \left( \frac{z-a}{b-a} + \frac{b-z}{b-a} \right) \\
& - \left( \frac{z-a}{b-a} \mu_{a \leq x \leq z} + \frac{b-z}{b-a} \mu_{z \leq x \leq b} \right)^2 \\
=& \frac{(z-a)(b-z)}{(b-a)^2} (\mu_{a \leq x \leq z} - \mu_{z \leq x \leq b})^2 .
\end{aligned}
\tag{20}
$$

$\square$

**Lemma 6.** *For any $a, b \in \mathbb{R}$, let $f(x)$ be a continuous and bounded function, and given $\int_a^b f(t) \, dt = 0$. Let $\mu_\Omega$ be the average of $f(x)$ in the area $C_\Omega$ satisfying the condition $\Omega$. For any $z^*$, existing a $z^\star$ such that*

$$\frac{(z^* - a)(b - z^*)}{(b-a)^2} (\mu_{a \leq x \leq z^*} - \mu_{z^* \leq x \leq b})^2 \leq \frac{|C_{f(x) \leq f(z^\star)}||C_{f(x) \geq f(z^\star)}|}{|C_{f(x)}|^2} (\mu_{f(x) \leq f(z^\star)} - \mu_{f(x) \geq f(z^\star)})^2 . \tag{21}$$

**Proof of Lemma 6.**   For any $z^*$, we let $z^\star$ be the value satisfying $\frac{|C_{f(x) \leq f(z^\star)}|}{|C_{f(x)}|} = \frac{z^* - a}{b - a}$ and $\frac{|C_{f(x) \geq f(z^\star)}|}{|C_{f(x)}|} = \frac{b - z^*}{b - a}$. Then we just need to prove that

$$\left( \mu_{a \leq x \leq z^*} - \mu_{z^* \leq x \leq b} \right)^2 \leq \left( \mu_{f(x) \leq f(z^\star)} - \mu_{f(x) \geq f(z^\star)} \right)^2 . \tag{22}$$

Essentially, this is to compare the inter-class variance between two child nodes after split in different dimensions ($x$ or $f(x)$). We denote by $Z = \int_{C_L} f(t) \, dt < 0$ the integral of $f(x)$ in the left node, then the inter-class variance equals to $(Z/p_{\text{left}} - (-Z)/(1 - p_{\text{left}}))^2 = Z^2/(p_{\text{left}} \cdot (1 - p_{\text{left}}))^2 \propto Z^2$. Because $p_{\text{left}}$ is fixed, we just need to consider $Z^2$. According to the rearrangement inequality, we know that the ordered sum is less than the disordered sum. Therefore, split along the dimension of $f(x)$, i.e., $Z_{f(x) \leq f(z^\star)}$ can be view as an ordered sum of $f(x)$, is smaller than the disorder sum $Z_{a \leq x \leq z^*}$. We have

$$Z_{f(x) \leq f(z^\star)} \leq Z_{a \leq x \leq z^*} \leq 0 , \tag{23}$$

so $Z_{f(x) \leq f(z^\star)}^2 \geq Z_{a \leq x \leq z^*}^2$. Since Eq. (22) is proved, then lemma is proved.   $\square$

**Lemma 7** (The distance between theoretical and empirical splits). *Assume that Assumption 1 is satisfied and the first-layer forest is consistent. Fix $\xi, \rho > 0$ and $k \in \mathbb{N}^*$. Then in the second layer, there exists $N \in \mathbb{N}^*$ such that, for all $n \geq N$,*

$$\Pr \left[ t_\infty \left( \hat{\boldsymbol{t}}_{k,n}(\boldsymbol{x}, \Theta), \mathcal{T}_k^*(\boldsymbol{x}, \Theta) \right) \leq \xi \right] \geq 1 - \rho . \tag{24}$$

**Proof of Lemma 7.**   We prove by induction that, for all $k$, with probability $1 - \rho$, for all $\xi > 0$ and all $n$ large enough,

$$d_\infty(\hat{\boldsymbol{t}}_{k,n}(\boldsymbol{x}, \Theta), \mathcal{T}_k^*(\boldsymbol{x}, \Theta)) \leq \xi . \tag{25}$$

Call this property $H_k$. Fix $k > 1$ and assume that $H_{k-1}$ is true. For all $\boldsymbol{t}_{k-1} \in \mathcal{T}_{k-1}(\boldsymbol{x})$, let

$$\hat{t}_{k,n}(\boldsymbol{t}_{k-1}) \in \underset{t_k}{\arg \min} \, \hat{L}_n(\boldsymbol{x}, \boldsymbol{t}_{k-1}, t_k) , \tag{26}$$

and

$$t_k^*(\boldsymbol{t}_{k-1}) \in \arg\min_{t_k} L^*(\boldsymbol{x}, \boldsymbol{t}_{k-1}, t_k) \, , \tag{27}$$

where the minimum is evaluated over $\{t_k \in \mathcal{S}_{C(\boldsymbol{x}, \boldsymbol{t}_{k-1})} \colon t_k^{(1)} \in \mathcal{M}_{\text{try}}\}$. Fix $\rho > 0$. In the rest of the proof, we assume that $\Theta$ is fixed.

$$d_\infty\left(\hat{t}_{k,n}(\hat{\boldsymbol{t}}_{k-1,n}), \mathcal{T}_k^*\right) \leq d_\infty\left(\hat{t}_{k,n}(\hat{\boldsymbol{t}}_{k-1,n}), t_k^*(\hat{\boldsymbol{t}}_{k-1,n})\right) + d_\infty\left(t_k^*(\hat{\boldsymbol{t}}_{k-1,n}), \mathcal{T}_k^*\right) \, . \tag{28}$$

According to Scornet et al. [29, Lemma 2 and preliminary result in Lemma 3], we have, with probability $\geq 1 - 2\rho$, for all $n$ large enough,

$$d_\infty\left(\hat{t}_{k,n}(\hat{\boldsymbol{t}}_{k-1,n}), t_k^*(\hat{\boldsymbol{t}}_{k-1,n})\right) \leq \xi \, . \tag{29}$$

Therefore, we just need to prove that $d_\infty\left(t_k^*(\hat{\boldsymbol{t}}_{k-1,n}), \mathcal{T}_k^*\right) \to 0$ in probability as $n \to \infty$. Let $\{\boldsymbol{t}_{k-1}^{*,i} \colon i \in \mathcal{I}\}$ be the set of best first $k-1$th theoretical splits and $t_k^*(\{\boldsymbol{t}_{k-1}^{*,i}\})$ be the $k$th theoretical spits given that the $k-1$ previous ones are $\boldsymbol{t}_{k-1}^{*,i}$.

Let

$$L^{i,*}(\boldsymbol{x}, t_k) = L_k^*(\boldsymbol{x}, \boldsymbol{t}_{k-1}^{*,i}, t_k) \quad \text{and} \quad \hat{L}^*(\boldsymbol{x}, t_k) = L_k^*(\boldsymbol{x}, \hat{\boldsymbol{t}}_{k-1,n}, t_k) \, , \tag{30}$$

$$t_k^*(\boldsymbol{t}_{k-1}^{*,i}) \in \arg\min_{t_k} L^{i,*}(\boldsymbol{x}, t_k) \quad \text{and} \quad t_k^*(\hat{\boldsymbol{t}}_{k-1,n}) \in \arg\min_{t_k} \hat{L}^{i,*}(\boldsymbol{x}, t_k) \, . \tag{31}$$

Then the original problem equals to that:

$$\inf_{i \in \mathcal{I}} d_\infty(t_k^*(\boldsymbol{t}_{k-1}^{*,i}), t_k^*(\hat{\boldsymbol{t}}_{k-1,n})) \to 0, \text{in probability, as } n \to \infty \, . \tag{32}$$

According to Scornet et al. [29, Technical Lemma 2], we just need to prove that, with probability $\geq 1 - \rho$,

$$\inf_i |L^{i,*}(\boldsymbol{x}, t_k^*(\hat{\boldsymbol{t}}_{k-1,n})) - L^{i,*}(\boldsymbol{x}, t_k^*(\boldsymbol{t}_{k-1}^{*,i}))| \leq 6\xi \, . \tag{33}$$

$$\inf_i |L^{i,*}(\boldsymbol{x}, t_k^*(\hat{\boldsymbol{t}}_{k-1,n})) - L^{i,*}(\boldsymbol{x}, t_k^*(\boldsymbol{t}_{k-1}^{*,i}))| \leq 2\inf_i \sup_{t_k} |\hat{L}^*(\boldsymbol{x}, t_k) - L^{i,*}(\boldsymbol{x}, t_k)|$$

$$\triangleright \text{ According to the continuity of } L_k^* \quad \leq 4\xi + 2\inf_i \sup_j |\hat{L}^*(\boldsymbol{x}, c_{j,\boldsymbol{x}}') - L^{i,*}(\boldsymbol{x}, c_{j,\boldsymbol{x}}')|$$

$$\triangleright \text{ According to Proposition (1.1)} \quad = 2\inf_i |L_k^*(\boldsymbol{x}, \hat{\boldsymbol{t}}_{k-1,n}, c_{d+1,\boldsymbol{x}}') - L_k^*(\boldsymbol{x}, \boldsymbol{t}_{k-1}^{*,i}, c_{d+1,\boldsymbol{x}}')|$$

$$+ 4\xi \, , \tag{34}$$

where $\mathcal{C}_{\delta,\boldsymbol{x}}' = \{c_{j,\boldsymbol{x}}' \colon 1 \leq j \leq d+1\}$ is a finite subset such that, for all $t_k$, $d_\infty(t_k, \mathcal{C}_{\delta,\boldsymbol{x}}') \leq \delta$, by default $d+1$-th dimension is the new feature. When the first-layer forest is consistent, the second-layer CART always theoretically split along the new feature. Since $L_k^*$ is uniformly continuous, by assumption $H_{k-1}$, $\inf_i \|\hat{\boldsymbol{t}}_{k-1,n} - \boldsymbol{t}_{k-1}^{*,i}\|_\infty \to 0$, we have

$$\inf_i |L^{i,*}(\boldsymbol{x}, t_k^*(\hat{\boldsymbol{t}}_{k-1,n})) - L^{i,*}(\boldsymbol{x}, t_k^*(\boldsymbol{t}_{k-1}^{*,i}))| \leq 6\xi \, . \tag{35}$$

The lemma is proved. □

### B.1  Proof of Proposition 1 and Proposition 2

**Proposition 1** (Priority of the new feature). *Assume the data set follows Assumption 1 and the first-layer forest is consistent. The following results hold for any CART in the second-layer forest.*

1. *In the infinite sample regime ($n = \infty$), we consider a single second-layer CART $h_\infty^{(2)}(\cdot, \Theta)$. All splits in this CART are performed along the new feature only.*

2. *In the finite sample regime ($n < \infty$), we consider a single second-layer CART $h_n^{(2)}(\cdot, \Theta)$. Fix $k \in \mathbb{N}^*$ and $\xi, \rho > 0$. Then, with probability $\geq 1 - \rho$, for all $n$ large enough, we have, the error of the first-layer forest is bound by $\xi$. As a consequence, the first $k$ splits $(j_{q,n}(\boldsymbol{x}), 1 \leq q \leq k)$ in this CART are performed along the new feature only.*

**Proof of Proposition 1.** First, we consider the infinite sample regime ($n = \infty$), which the first-layer forest estimation is precise, i.e., $h_{M,\infty}(\boldsymbol{x}) = f(\boldsymbol{x})$.

Recall that, for any cell $A$, the empirical CART criterion used to split $A$ in the random forest is defined in Eq.(7). For any split $(j, z)$, we denote the theoretical version of $L^*(j, z)$ by

$$L^*(j, z) = \text{Var}[y|\boldsymbol{x} \in A] - \Pr[x^{(j)} < z] \, \text{Var}[y|x^{(j)} < z \wedge \boldsymbol{x} \in A] \tag{36}$$
$$- \Pr[x^{(j)} \geq z] \, \text{Var}[y|x^{(j)} \geq z \wedge \boldsymbol{x} \in A] \, .$$

According to the strong law of large numbers, we have $L_n(j, z) \to L^*(j, z)$ almost surely as $n \to \infty$ for all splits $(j, z) \in \mathcal{S}_A$. Thus we have the best theoretical split $(j^*, z^*)$ of the cell $A$

$$(j^*, z^*) \in \underset{\substack{(j^*, z^*) \in \mathcal{S}_A \\ j \in \mathcal{M}_{try}}}{\arg\max} \; L^*(j, z) \, . \tag{37}$$

In the random forest and deep forest, $\mathcal{M}_{try}$ is also an important parameter. Unlike random forests, where we give all features the same probability to be selected, deep forests often choose new features with higher probability or even make new features mandatory in order to make use of the representation information brought by new features. Therefore, the deep forest or CART analyzed in this paper selects new features by default. Otherwise, the tree and forest in the second layer will be equivalent to the random forest, because it does not inherit any information in the first layer.

**1. Infinite sample region ($n = \infty$)**

We start by proving the result in dimension $d = 1$. Letting $C_x = [a, b]$ be any cell, and recalling that $y = f(x^{(1)}) + \epsilon$, then in the infinite sample regime we define the theoretical version of CART's split criterion function on the raw feature as

$$L^*(1, z) = \text{Var}[y|x^{(1)} \in C_x] - \Pr[a \leq x^{(1)} \leq z|x^{(1)} \in C_x] \, \text{Var}[y|a \leq x^{(1)} \leq z]$$
$$- \Pr[z \leq x^{(1)} \leq b|x^{(1)} \in C_x] \, \text{Var}[y|z \leq x^{(1)} \leq b]$$
$$= -\frac{1}{(b-a)^2} \left( \int_a^b f(t) \, \mathrm{d}t \right)^2 + \frac{1}{(b-a)(z-a)} \left( \int_a^z f(t) \, \mathrm{d}t \right)^2$$
$$+ \frac{1}{(b-a)(b-z)} \left( \int_z^b f(t) \, \mathrm{d}t \right)^2 \, .$$

Let $I = \int_a^b f(t) \, \mathrm{d}t$ and $F(z) = \int_a^z f(t) \, \mathrm{d}t$. Then, the theoretical criterion function

$$L^*(1, z) = \frac{1}{(z-a)(b-z)} \left( F(z) - I \cdot \frac{z-a}{b-a} \right)^2 \, . \tag{38}$$

According to the consistency of original random forest [29], we have the new feature $h_\infty^{(1)}(x) = f(x)$, which is a consistent estimation of the target function. Thus the theoretical criterion function on the new feature takes the form

$$L^*(h, z) = \text{Var}[y|h \in C_h] - \Pr[h \leq z|h \in C_h] \, \text{Var}[y|h \leq z]$$
$$- \Pr[h \geq z|h \in C_h] \, \text{Var}[y|h \geq z]$$
$$= -\frac{1}{|C_h|^2} \left( \int_{C_h} f(t) \, \mathrm{d}t \right)^2 + \frac{1}{|C_{h \leq z}||C_h|} \left( \int_{C_{h \leq z}} f(t) \, \mathrm{d}t \right)^2$$
$$+ \frac{1}{|C_{h \geq z}||C_h|} \left( \int_{C_{h \geq z}} f(t) \, \mathrm{d}t \right)^2$$
$$= -\frac{1}{|C_f|^2} \left( \int_{C_f} f(t) \, \mathrm{d}t \right)^2 + \frac{1}{|C_{f \leq z}||C_f|} \left( \int_{C_{f \leq z}} f(t) \, \mathrm{d}t \right)^2$$
$$+ \frac{1}{|C_{f \geq z}||C_f|} \left( \int_{C_{f \geq z}} f(t) \, \mathrm{d}t \right)^2$$

Let $I = \int_{C_f} f(t)\,\mathrm{d}t = \int_a^b f(t)\,\mathrm{d}t$ and $G(z) = \int_{C_{f \leq z}} f(t)\,\mathrm{d}t$. Then, the theoretical criterion function becomes

$$L^*(h, z) = \frac{1}{|C_{h \geq z}||C_{h \leq z}|}\left(G(z) - I \cdot \frac{|C_{h \leq z}|}{|C_h|}\right)^2 . \tag{39}$$

Without loss of generality, we let $I = 0$, then the optimal split along the raw feature is

$$z^\star = \arg\max_{z \in [a,b]} L^*(1, z) = \arg\max_{z \in [a,b]} \frac{1}{(z-a)(b-z)}G^2(z) . \tag{40}$$

We compare the maximum value of the CART-split criterion along the raw feature and the new feature:

$$
\begin{aligned}
\max_z L^*(1, z) &= \frac{1}{(z^*-a)(b-z^*)}F^2(z^*)\\
\triangleright \text{ According to Lemma } 5 \quad &= \frac{(z^*-a)(b-z^*)}{(b-a)^2}(\mu_{a \leq x \leq z^*} - \mu_{z^* \leq x \leq b})^2\\
\triangleright \text{ According to Lemma } 6 \quad &\leq \frac{|C_{h \leq f(z^\star)}||C_{h \geq f(z^\star)}|}{|C_h|^2}(\mu_{h \leq f(z^\star)} - \mu_{h \geq f(z^\star)})^2\\
\triangleright \text{ According to Lemma } 5 \quad &= \frac{1}{|C_{h \leq f(z^\star)}||C_{h \geq f(z^\star)}|}G^2(f(z^\star))\\
&\leq \max_z L^*(h, z) ,
\end{aligned}
\tag{41}
$$

and the $d = 1$ case is proved.

Next, we consider the $d > 1$ case, where $A = \prod_{j=1}^d [a_j, b_j] \subset [0, 1]^d$. We know that, for all $1 \leq j \leq d$, there exists a constant $I$ such that

$$
\begin{aligned}
&\int_{a_1}^{b_1} \cdots \int_{a_d}^{b_d} f(\boldsymbol{x})\,\mathrm{d}x^{(1)} \ldots x^{(j-1)}x^{(j+1)} \ldots x^{(d)}\\
&= f_j(x^{(j)}) + \int_{a_1}^{b_1} \cdots \int_{a_d}^{b_d} \sum_{\ell \neq j} f_\ell(x^{(\ell)})\,\mathrm{d}x^{(1)} \ldots x^{(j-1)}x^{(j+1)} ,
\end{aligned}
\tag{42}
$$

which can be simply denoted as

$$\int_{C_{x^{(-j)}}} f(\boldsymbol{x})\,\mathrm{d}\boldsymbol{x}^{(-j)} = f_j(x^{(j)}) + \int_{C_{x^{(-j)}}} \sum_{\ell \neq j} f_\ell(x^{(\ell)})\,\mathrm{d}\boldsymbol{x}^{(-j)} . \tag{43}$$

Let $I_j = \int_{C_{x^{(-j)}}} \sum_{\ell \neq j} f_\ell(x^{(\ell)})\,\mathrm{d}\boldsymbol{x}^{(-j)}$, which does not depend on $x^{(j)}$. Since $f(\cdot)$ is an additive model, for all $j$ and all $x^{(j)}$,

$$\int_{a_j}^{z_j} \int_{C_{x^{(-j)}}} f_j(\boldsymbol{x})\,\mathrm{d}\boldsymbol{x}^{(-j)}\,\mathrm{d}x^{(j)} = \int_{a_j}^{z_j} f(x^{(j)})\,\mathrm{d}x^{(j)} + (z_j - a_j)I_j . \tag{44}$$

Let $z_j = z$, $a_j = a$, $b_j = b$, the theoretical criterion function on the raw feature takes the form

$$
\begin{aligned}
L^*(j, z) =\ & \mathrm{Var}[y|x^{(j)} \in C_x] - \Pr[a \leq x^{(j)} \leq z|x^{(j)} \in C_x]\,\mathrm{Var}[y|a \leq x^{(j)} \leq z]\\
& - \Pr[z \leq x^{(j)} \leq b|x^{(j)} \in C_x]\,\mathrm{Var}[y|z \leq x^{(j)} \leq b]\\
=\ & -\frac{1}{(b-a)^2}\left(\int_{C_{x^{(-j)}}} \int_a^b f(x^{(j)})\,\mathrm{d}x^{(j)}\,\mathrm{d}\boldsymbol{x}^{(-j)}\right)^2\\
& + \frac{1}{(b-a)(z-a)}\left(\int_{C_{x^{(-j)}}} \int_a^z f(x^{(j)})\,\mathrm{d}x^{(j)}\,\mathrm{d}\boldsymbol{x}^{(-j)}\right)^2\\
& + \frac{1}{(b-a)(b-z)}\left(\int_{C_{x^{(-j)}}} \int_z^b f(x^{(j)})\,\mathrm{d}x^{(j)}\,\mathrm{d}\boldsymbol{x}^{(-j)}\right)^2\\
=\ & \frac{1}{(z-a)(b-z)}\left(F(z) - I \cdot \frac{z-a}{b-a}\right)^2 + \frac{(b-a)^2 + (b-z)^2 + (z-a)^2}{2(b-a)^2(b-z)(z-a)}I_j
\end{aligned}
$$

According to the proof of $d = 1$, we can fix $z - a$ and $b - z$. Then we just need consider $\frac{1}{(z-a)(b-z)} \left( F(z) - I \cdot \frac{z-a}{b-a} \right)^2$, which is same as $d = 1$.

Intuitively, in the multi-dimensional case, the correlation between $y$ and $\boldsymbol{x}$ is scattered to each dimension due to the assumption that each dimension is independent and $y = f(\boldsymbol{x}) + \epsilon$ is an additive model. Therefore, when splitting is calculated separately in each dimension, the gain is not as large as that caused by the only dimension splitting in the case of one dimension $L^*(1, z)$ above. According to the result of $d = 1$, we have

$$\max_{j,z} L^*(j, z) < \max_z L^*(h, z) , \tag{45}$$

and the $d > 1$ case is proved.

**2. Finite sample region ($n < \infty$)**

Fix $k \in \mathbb{N}^*$ and $\rho, \xi > 0$. According to Lemma 7, with probability $1 - \rho$, for all $n$ large enough, there exists a sequence of theoretical first $k$ splits $\boldsymbol{t}_k^*(\boldsymbol{x}, \Theta)$ such that

$$d_\infty \left( \boldsymbol{t}_k^*(\boldsymbol{x}, \Theta), \hat{\boldsymbol{t}}_{k,n}(\boldsymbol{x}, \Theta) \right) \le \xi . \tag{46}$$

Therefore, with probability $\ge 1 - \rho$, for all $n$ large enough and all $1 \le j \le k$, the $j$-th empirical split $\hat{t}_{j,n}^*(\boldsymbol{x}, \Theta)$ is performed along the same dimension as $t_j^*(\boldsymbol{x}, \Theta)$. According to the result of theoretical criterion splits, the splits are always performed along the new features, which is the most informative variable. Consequently, for all $\boldsymbol{x}, \Theta$ and for all $1 \le j \le k$, each empirical split $\hat{t}_{j,n}^*(\boldsymbol{x}, \Theta)$ is performed along the new features only. Then the proposition is proved. $\square$

**Proposition 2** (Variation of $f$ in the empirical cell). *Assume that Assumption 1 holds and the first-layer forest is consistent. The following results hold for any CART in the second-layer forest $h_n^{(2)}(\cdot, \Theta)$. After splitting along the new feature, the CART will estimate the residual of the first-layer forest estimation. Then for all $\rho, \xi > 0$, there exists $N \in \mathbb{N}^*$ such that, for all $n > N$,*

$$\Pr \left[ \Delta \left( f, C_{\Pi,n}(\boldsymbol{x}, \Theta) \right) \le \xi \right] \ge 1 - \rho . \tag{11}$$

**Proof of Proposition 2.** According to Proposition 1, we know that in the theoretical version ($n = \infty$), the CART in the second layer will always split along the new feature. Since $f(\boldsymbol{x})$ is uniformly continuous, the result is clear if $\mathrm{diam}(C_k^*(\boldsymbol{x}, \Theta))$ tends to 0 as $k$ tends to infinity. Thus, in the following proof, we assume that $\mathrm{diam}(C_k^*(\boldsymbol{x}, \Theta))$ does not tend to 0. We denote by $h$ the new feature dimension. $(C_k^*(\boldsymbol{x}, \Theta))$ is a decreasing sequence of compact sets, there exist $a_\infty(h, \Theta)$ and $b_\infty(h, \Theta)$ such that

$$C_\infty^*(\boldsymbol{x}, \Theta) \triangleq \cap_{k=1}^\infty C_k^*(\boldsymbol{x}, \Theta) = [a_\infty(h, \Theta), b_\infty(h, \Theta)] . \tag{47}$$

Since $\mathrm{diam}(C_k^*(\boldsymbol{x}, \Theta))$ does not tend to zero, we have $a_\infty(h, \Theta) < b_\infty(h, \Theta)$. If the criterion $L^*$ is identically zero for all cuts $z$ in $C_\infty^*(\boldsymbol{x}, \Theta)$ then recall Eq. (39), we have

$$G(z) \propto \frac{z - a}{b - a} . \tag{48}$$

This proves that $G(z)$ is linear in $z$, so $f$ is a constant on $[a, b]$. This implies that $\Delta(f, C_\infty^*(\boldsymbol{x}, \Theta)) = 0$. Since $f$ is uniformly continuous,

$$\lim_{k \to \infty} \Delta(f, C_k^*(\boldsymbol{x}, \Theta)) = \Delta(f, C_\infty^*(\boldsymbol{x}, \Theta)) = 0 . \tag{49}$$

Fix $\xi, \rho > 0$. Since almost sure convergence implies convergence in probability, according to the result above, there exists $k_0 \in \mathbb{N}^*$ such that

$$\Pr[\Delta(f, C_{k_0}^*(\boldsymbol{x}, \Theta) \le \xi] \ge 1 - \rho . \tag{50}$$

According to Lemma 7, for all $\xi' > 0$, there exists $N \in \mathbb{N}^*$ such that, for all $n \ge N$,

$$\Pr[d_\infty(\hat{\boldsymbol{t}}_{k_0,n}(\boldsymbol{x}, \Theta), \mathcal{T}_{k_0}^*(\boldsymbol{x}, \Theta)) \le \xi'] \ge 1 - \rho . \tag{51}$$

since $f$ is uniformly continuous, we set $\xi'$ sufficiently small such that, for all $\boldsymbol{x} \in [0, 1]^d$, for all $\boldsymbol{t}_{k_0}, \boldsymbol{t}_{k_0}'$ satisfying $d_\infty(\boldsymbol{t}_{k_0}, \boldsymbol{t}_{k_0}') \le \xi'$, we have

$$|\Delta(f, C(\boldsymbol{x}, \boldsymbol{t}_{k_0}) - \Delta(f, C(\boldsymbol{x}, \boldsymbol{t}_{k_0}')| \le \xi . \tag{52}$$

Combining Eq. (51) and (52), we obtain

$$\Pr[|\Delta(f, C_{k_0,n}(\boldsymbol{x}, \Theta)) - \Delta(f, C_{k_0}^*(\boldsymbol{x}, \Theta))| \leq \xi] \geq 1 - \rho . \tag{53}$$

Then we can obtain the result from $\Delta(f, C) \leq \Delta(f, C')$ whenever $C \subset C'$,

$$\Pr[\Delta(f, C_{\Pi,n}(\boldsymbol{x}, \Theta)) \leq \xi] \geq 1 - 2\rho . \tag{54}$$

$\square$

## B.2   Proof of Theorem 3

**Theorem 3** (Universal consistency). *Let $M \geq 1$. Consider two-layer deep forest $\bar{h}_{2M,n}$ given by Eq. (6) and Breiman's random forest $h_{M,n}$ given by Eq. (4) for the random CARTs satisfying $a_n \to \infty, t_n \to \infty$ and $t_n(\log a_n)^9/a_n \to 0$. Then under the setting described in Section 3 and assume the data set follows Assumption 1,*

1. *[29, Theorem 1] the Breiman's random forest $h_{2M,n}$ is consistent for any $M \geq 1$,*

2. *the two-layer deep forest $\bar{h}_{M,n}$ is consistent for any $M \geq 1$.*

**Proof of Theorem 3.**   **(T3.1).** The universal consistency of Breiman's random forest is proved by Scornet et al. [29, Thoerem 1].

**(T3.2).** Similar as Scornet et al. [29, Theorem 1], we can use the bounded variation of $f$ in the empirical cell in Proposition 2 to control the approximation error. Let $\mathcal{H}_n(\Theta)$ be the set of all functions $h : [0,1]^{d+1} \to \mathbb{R}$ piecewise constant on each cell of the partition $\Pi_n(\Theta)$. Thus the second-layer CART estimator $h_n^{(2)}(\boldsymbol{x}, \Theta)$ satisfies

$$h_n^{(2)}(\boldsymbol{x}, \Theta) \in \arg\min_{h \in \mathcal{H}_n(\Theta)} \frac{1}{a_n} \sum_{i \in \mathcal{I}_{n,\Theta}} |h([\boldsymbol{x}_i, h_{M,n}^{(1)}(\boldsymbol{x}_i)]) - y_i|^2 . \tag{55}$$

Let $(\beta_n)_n$ be a positive sequence, and define the truncated operator $T_{\beta_n}$ by

$$\begin{cases} T_{\beta_n} u = u, & \text{if } |u| < \beta_n , \\ T_{\beta_n} u = \text{sign}(u)\beta_n, & \text{if } |u| \geq \beta_n . \end{cases} \tag{56}$$

Then, we define $T_{\beta_n} h_n^{(2)}(\boldsymbol{x}, \Theta)$, $y_L = T_L y$ and $y_{i,L} = T_L y_i$.

For all $n$ large enough, we have

$$\begin{aligned} \mathbb{E} \inf_{\substack{h \in \mathcal{H}_n(\Theta) \\ \|h\|_\infty \leq \beta_n}} \mathbb{E}_{\boldsymbol{x}}[h(\boldsymbol{x}) - f(\boldsymbol{x})]^2 &\leq \mathbb{E} \inf_{\substack{h \in \mathcal{H}_n(\Theta) \\ \|h\|_\infty \leq \|f\|_\infty}} \mathbb{E}_{\boldsymbol{x}}[h(\boldsymbol{x}) - f(\boldsymbol{x})]^2 \\ &\leq \mathbb{E}[\Delta(f, C_{\Pi,n}(\boldsymbol{x}, \Theta))]^2 \\ &\leq \xi^2 + 4\|f\|_\infty^2 \Pr[\Delta(f, C_{\Pi,n}(\boldsymbol{x}, \Theta)) \geq \xi] . \end{aligned} \tag{57}$$

Connecting with Proposition 2, we have

$$\mathbb{E} \inf_{\substack{h \in \mathcal{H}_n(\Theta) \\ \|h\|_\infty \leq \beta_n}} \mathbb{E}_{\boldsymbol{x}}[h(\boldsymbol{x}) - f(\boldsymbol{x})]^2 \leq 2\xi^2 . \tag{58}$$

This proves that the approximation error tends to zero.

The proof of estimation error and untruncated estimate is same as Scornet et al. [29, Thoerem 1]. The parameter $t_n$ allows us to control the size of the leaves of CART, which allows us to have enough samples in each leaf node to smooth the impact of noise, so as to control the estimation error.

$$\Pr\left[\sup_{\substack{h \in \mathcal{H}_n(\Theta) \\ \|h\|_\infty \leq \beta_n}} \left| \frac{1}{a_n} \sum_{i \in \mathcal{I}_{n,\Theta}} [h(\boldsymbol{x}_i) - y_{i,L}]^2 - \mathbb{E}[h(\boldsymbol{x}) - y_L]^2 \right| \geq \xi \right] \leq 8\exp\left(-\frac{a_n C_{\xi,n}}{\beta_n^4}\right) , \tag{59}$$

where

$$C_{\xi,n} = \frac{\xi^2}{2048} + \mathcal{O}\left(\frac{t_n(\log a_n)^9}{a_n}\right) . \tag{60}$$

According to the condition $t_n(\log a_n)^9/a_n \to 0$, we have $C_{\xi,n} \to \frac{\xi^2}{2048}$. Then, we can bound the estimation error as follow

$$\mathbb{E}\left[\sup_{\substack{h\in\mathcal{H}_n(\Theta)\\ \|h\|_\infty\le\beta_n}}\left|\frac{1}{a_n}\sum_{i\in\mathcal{I}_{n,\Theta}}[h(\boldsymbol{x}_i)-y_{i,L}]^2-\mathbb{E}[h(\boldsymbol{x})-y_L]^2\right|\right] \le \xi + 16(\beta_n+L)^2\exp\left(-\frac{a_nC_{\xi,n}}{\beta_n^4}\right)$$
$$\le 2\xi .$$
(61)

This proves that the estimation error tends to zero. Connecting the approximation and estimation error together with Györfi et al. [46, Theorem 10.2], the consistency of a CART of second-layer deep forest is proved.

The universal consistency can be proved via Biau et al. [25, Proposition 1], which guarantees that the error of forest estimator is no more than twice that of individual randomized CART. □

### B.3 Proof of Theorem 4

**Theorem 4** (Depth is more powerful than width). *Let $M \ge 1$. Consider two-layer deep forest $\bar{h}_{2M,n}$ given by Eq. (6) and Breiman's random forest $h_{M,n}$ given by Eq. (4) for the random CARTs satisfying $a_n \to \infty, t_n \to \infty, t_n = a_n$ and $a_n \log n/n \to 0$. Then under the setting described in Section 3 and assume the data set follows Assumption 1 and 2, the following results hold*

1. *[29, Theorem 2] [30, Theorem 3] The Breiman's random forest $h_{\infty,n}$ is consistent, and for all $M, n \in \mathbb{N}$,*

$$0 \le R(h_{2M,n}) - R(h_{\infty,n}) \le \frac{8\|f\|_\infty^2 + 8\sigma^2(1+4\log n)}{M} .$$
(16)

2. *The two-layer deep forest $\bar{h}_{\infty,n}$ is consistent, and for all $M, n \in \mathbb{N}$, if $\Delta(f, C_{\Pi,n}(\boldsymbol{x},\Theta))$ is small enough, then*

$$0 \le R(\bar{h}_{2M,n}) - R(\bar{h}_{\infty,n}) \le \frac{64\|f\|_\infty^2 + 64\sigma^2(1+4\log n)}{M^2} .$$
(17)

**Proof of Theorem 4.** **(T4.1).** The consistency of the infinite Breiman's random forest is proved by Scornet et al. [29]. And the convergence rate of the finite random forest with the number of trees $M$ is proved by Scornet [30].

**(T4.2).** Because each cell contains only one sample in this regime, we define

$$W_{ni}(\boldsymbol{x}) = \mathbb{E}_\Theta[\mathbb{1}_{\boldsymbol{x}_i\in C_{\Pi,n}(\boldsymbol{x},\Theta)}] ,$$
(62)

the infinite two layer deep forest estimation can rewriten as

$$\bar{h}_{\infty,n}(\boldsymbol{x}) = h_{\infty,n}^{(2)}([\boldsymbol{x}, h_{\infty,n}^{(1)}(\boldsymbol{x})]) = \sum_{i=1}^n W_{ni}([\boldsymbol{x}, h_{M,n}^{(1)}(\boldsymbol{x})])y_i .$$
(63)

Thus,

$$\mathbb{E}[\bar{h}_{\infty,n}(\boldsymbol{x}) - f(\boldsymbol{x})] \le 2\mathbb{E}\left[\sum_{i=1}^n W_{ni}([\boldsymbol{x}, h_{\infty,n}^{(1)}(\boldsymbol{x})])(y_i - f(\boldsymbol{x}_i))\right]^2$$
$$+ 2\mathbb{E}\left[\sum_{i=1}^n W_{ni}([\boldsymbol{x}, h_{\infty,n}^{(1)}(\boldsymbol{x})])(f(\boldsymbol{x}_i) - f(\boldsymbol{x}))\right]^2$$
$$\triangleq 2I_n + 2J_n .$$
(64)

Similar as Scornet et al. [29], we recall Proposition 2 to control the approximation error of the two-layer deep forest.

$$J_n \le \mathbb{E}\left[\sum_{i=1}^n \mathbb{1}_{[\boldsymbol{x}_i, h_{\infty,n}^{(1)}(\boldsymbol{x}_1)]\in C_{\Pi,n}([\boldsymbol{x}, h_{\infty,n}^{(1)}(\boldsymbol{x})],\Theta)}\Delta^2(f, C_{\Pi,n}([\boldsymbol{x}, h_{\infty,n}^{(1)}(\boldsymbol{x})],\Theta))\right]$$
$$\le \mathbb{E}[\Delta^2(f, C_{\Pi,n}([\boldsymbol{x}, h_{\infty,n}^{(1)}(\boldsymbol{x})],\Theta))]$$
$$\le \xi(4\|f\|_\infty^2 + 1) \qquad \triangleright \quad \text{According to Proposition 2.}$$
(65)

The proof of estimation error is same as Scornet et al. [29, Thoerem 2]. the estimation error is controlled by forcing the subsampling rate $a_n/n$ to be $o(1/\log n)$. For simplification, we denote $[\boldsymbol{x}, h_{\infty,n}^{(1)}(\boldsymbol{x})]$ as $\mathbf{X}$

$$
\begin{aligned}
I_n &= \mathbb{E}\left[\sum_{i,j=1}^n W_{ni}(\mathbf{X})W_{nj}(\mathbf{X})\left(y_i - f\left(\boldsymbol{x}_i\right)\right)\left(y_j - f\left(\boldsymbol{x}_j\right)\right)\right] \\
&= \mathbb{E}\left[\sum_{i=1} W_{ni}^2(\mathbf{X})\left(y_i - f\left(\boldsymbol{x}_i\right)\right)^2\right] + I_n' \,,
\end{aligned}
\tag{66}
$$

where

$$
I_n' = \mathbb{E}\left[\sum_{\substack{i,j \\ i\neq j}} \mathbb{1}_{\mathbf{X}\overset{\Theta}{\leftrightarrow}\mathbf{X}_i} \mathbb{1}_{\mathbf{X}\overset{\Theta'}{\leftrightarrow}\mathbf{X}_j}\left(y_i - f\left(\boldsymbol{x}_i\right)\right)\left(y_j - f\left(\boldsymbol{x}_j\right)\right)\right] \,.
\tag{67}
$$

By Assumption 2 and Scornet et al. [29, Lemma 4], for all $n$ large enough, $|I_n'| \le \xi$. Then,

$$
\begin{aligned}
|I_n| &\le \xi + \mathbb{E}\left[\max_{1\le\ell\le n} W_{n\ell}(\mathbf{X}) \max_{1\le i\le n} \varepsilon_i^2\right] \\
&\le \xi + \max_{1\le i\le n} \Pr_{\Theta}\left[\mathbf{X}\overset{\Theta}{\leftrightarrow}\mathbf{X}_i\right] \mathbb{E}\left[\max_{1\le i\le n} \varepsilon_i^2\right] \\
&\le \xi + \frac{a_n}{n}\mathbb{E}\left[\max_{1\le i\le n} \varepsilon_i^2\right] \\
&\le \xi + C\frac{a_n \log n}{n} \le 2\xi \,. \qquad \triangleright \text{ According to } a_n/n \sim o(1/\log n) \,.
\end{aligned}
\tag{68}
$$

Connecting the approximation and estimation error together, the consistency of an infinite two layer deep forest is proved.

Different from the bagging-style mechanism in random forest, the residual-style mechanism shown in Proposition 2 makes the second-layer forest in DF can reuse the first-layer estimation and focus on the residual learning. According to Scornet [30, Theorem 3.3], we have

$$
R(h_{M,n}) - R(h_{\infty,n}) \le \frac{8}{M} \times \left(\|f\|_\infty^2 + \sigma^2(1 + 4\log n)\right) \,,
\tag{69}
$$

for the first-layer forest estimation. When $n$ is large enough, we have $R(h_{\infty,n}) < \xi$ and

$$
R(h_{M,n}) \le \xi + \frac{8}{M} \times \left(\|f\|_\infty^2 + \sigma^2(1 + 4\log n)\right) \,.
\tag{70}
$$

According to Proposition 1, the first $k$ splits are only along the new feature dimension. This is equivalent to using a piecewise constant function of $h_{M,n}^{(1)}$ to copy the first-layer forest estimation, which is independent of $\Theta$. After the size of piece is small than the first-layer error, the raw features are used to estimate the residual $r(\boldsymbol{x}) = f(\boldsymbol{x}) - h_{M,n}^{(1)}(\boldsymbol{x})$. Thus, we obtain the bound for residual :

$$
\|r\|_\infty^2 \le \frac{8}{M} \times \left(\|f\|_\infty^2 + \sigma^2(1 + 4\log n)\right) \,.
\tag{71}
$$

As for the noise of the residual, we first consider the $R(h_{M,n}^{(1)}) \ge \mathbb{E}[\epsilon^2] = \sigma^2$ case: The first-layer estimator is too weak to filter noise, so the noise of the residual is still $\epsilon$. Next, we consider the $R(h_{M,n}^{(1)}) < \mathbb{E}[\epsilon^2] = \sigma^2$ case: Since the first-layer estimator smoothes part of the noise, the noise $\epsilon'$ is reduced in the residual. When $n$ is large enough, the size of noise can be bounded by the variation of $f$ in the empirical cell of the first-layer forest

$$
\mathbb{E}\epsilon'^2 \le c\Delta^2(f, C_{\Pi,n}(\boldsymbol{x}, \Theta)) \,.
\tag{72}
$$

Then we obtain the risk of the finite second-layer forest,

$$
\begin{aligned}
R(h_{M,n}^{(2)}) =& \xi + \frac{1}{M} \times \mathbb{E}\left[\mathrm{Var}_\Theta\left[\sum_{i=1}^n W_{ni}(\boldsymbol{x},\Theta)\left(r(\boldsymbol{x}_i)+\varepsilon_i'\right)\right]\right] \\
\leq& \xi + \frac{1}{M} \times \left[8\|r\|_\infty^2 + 2\mathbb{E}\left[\mathbb{V}_\Theta\left[\sum_{i=1}^n W_{ni}(\boldsymbol{x},\Theta)\varepsilon_i'\right]\right]\right] \\
\leq& \xi + \frac{1}{M} \times \left[8\|r\|_\infty^2 + 8\sigma'^2\mathbb{E}\left[\max_{1\leq i\leq n}\frac{\varepsilon_i'}{\sigma'}\right]^2\right] \\
\leq& \xi + \frac{64\|f\|_\infty^2 + 64\sigma^2(1+4\log n)}{M^2} + \frac{c\Delta^2(f,C_{\Pi,n}(\boldsymbol{x},\Theta)}{M} \ .
\end{aligned}
\tag{73}
$$

Thus, if the variation of $f$ in the empirical cell is small enough, then the two layer deep forest can obtain a faster convergence rate *w.r.t.* $M$. This theorem is proved. □

## C  Additional Results for Simulation Experiments

The experimental setting is the same as in Section 7, except that we set $f(\boldsymbol{x})$ to be a nonlinear function, that is,

$$
f(\boldsymbol{x}) = \frac{1}{5}\left(\sin 2\pi x_1 + \cos 2\pi x_2 + \sin(2\pi x_3 + \pi/3) + \cos(2\pi x_4 + \pi/3) + \sin 6\pi x_5\right) \ . \tag{74}
$$

As plotted in Figure S5, we also observe the same tendency as in Section 7.2, that using inconsistent trees is better in practice, and the 2-layer deep forest (DF) convergences faster *w.r.t.* the number of trees $M$. We also check the effect depth as in Section 7.1, and Figure S4 convinces us that the new feature has priority in splitting. Furthermore, we set $f(\boldsymbol{x})$ to be an interacted function,

$$
f(\boldsymbol{x}) = \frac{1}{5}\left(x_1 + x_2 x_3 + x_2^2 x_3^{1/2} + x_3 \sin 2\pi x_4 + \sin(2\pi x_4)\cos(6\pi x_5 + \pi/4)\right) \ . \tag{75}
$$

As plotted in Figure S7, using inconsistent trees is better in practice, and the 2-layer deep forest (DF) convergences faster *w.r.t.* the number of trees $M$. Figure S6 convinces us that the new feature has priority in splitting.

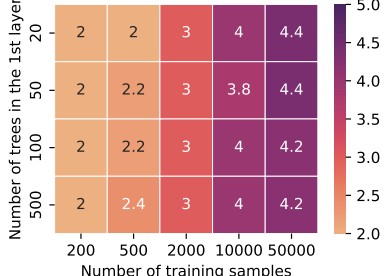

(a) Heatmap of the effective depth of new feature under different settings of the first layer forest.

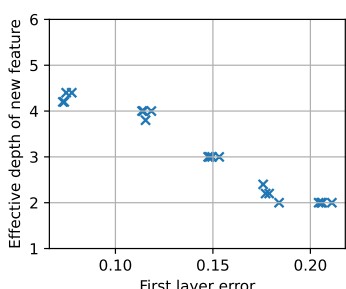

(b) Effective depth of new feature against the predictive error of the first layer.

Figure S4: Illustrations of the effective depth of the new feature (the consecutive levels from root node that split on the new feature only). The larger the effective depth, the higher priority the new feature takes in being chosen as the split feature under the CART-split criterion.

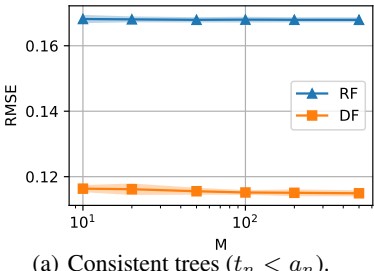
(a) Consistent trees ($t_n < a_n$).

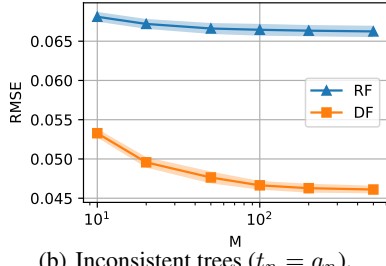
(b) Inconsistent trees ($t_n = a_n$).

Figure S5: Root mean square error with the increasing of number of trees $M$.

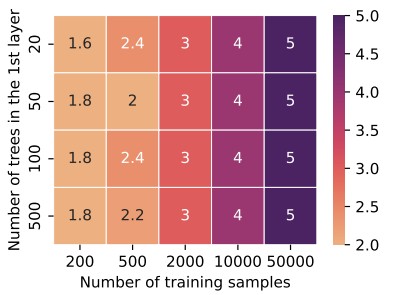
(a) Heatmap of the effective depth of new feature under different settings of the first layer forest.

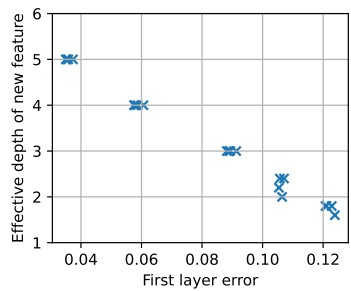
(b) Effective depth of new feature against the predictive error of the first layer.

Figure S6: Illustrations of the effective depth of the new feature (the consecutive levels from root node that split on the new feature only). The larger the effective depth, the higher priority the new feature takes in being chosen as the split feature under the CART-split criterion.

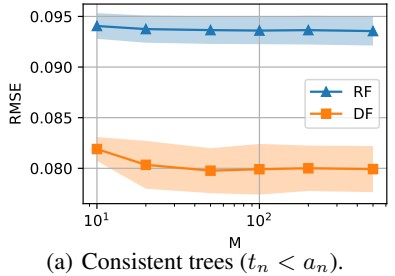
(a) Consistent trees ($t_n < a_n$).

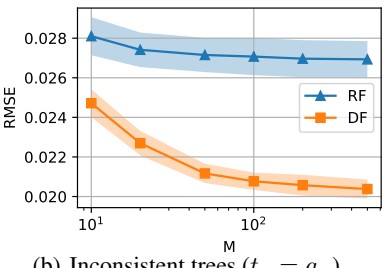
(b) Inconsistent trees ($t_n = a_n$).

Figure S7: Root mean square error with the increasing of number of trees $M$.

# D    Results on real world regression problems

In this section, we conduct experiments on real-world data sets. It should be noted that when we do experiments on real-world data sets, the number of samples is finite and the true underlying function is unknown, so there is gap from the theoretical analysis of consistency. Even so, the generalization performance on real data sets still shows a tendency for 2-layer deep forest to be more efficient than 1-layer random forest.

**Data sets.**    We conduct experiments on 3 real-world regression problems and the detail statistics of the data sets are shown in Table S2.

1. `housing data set`: This dataset contains information collected by the U.S Census Service concerning housing in the area of Boston Mass. It was obtained from the StatLib archive[1], and has been used extensively throughout the literature to benchmark algorithms. However, these comparisons were primarily done outside of Delve and are thus somewhat suspect.

2. `cadata data set`: This data set gathers information on housing prices using all neighborhood groups in California from the 1990 census. It calculates the distance between the centroids of each block group measured in latitude and longitude. It excludes all block groups reporting zero entries for the independent and dependent variables.

3. `acoustic data set`: This data set is collected from simulation result of COMSOL platform. It aims at predicting the energy focusing effect of an acoustic system based on 21 angle parameters.

| Data set | # of samples | # of features |
|----------|--------------|---------------|
| housing  | 506          | 13            |
| cadata   | 20,640       | 8             |
| acoustic | 4,000        | 21            |

Table S2: The average test error measured by RMSE of 5 runs on benchmark data sets. DF is better than RF in test error.

**Generalization performance.**    $M$ is set to 500 and the trees are fully grown as is commonly used in the literature. The average RMSE on test set of 5 runs is reported in Table S3.

| Data set | RF    | DF        |
|----------|-------|-----------|
| housing  | 3.62  | **3.56**  |
| cadata   | 50208 | **49363** |
| acoustic | 2.47  | **2.34**  |

Table S3: The average test error measured by RMSE of 5 runs on benchmark data sets. DF is better than RF in test error.

**Priority of new features.**    We vary the number of training samples from 10% to 100% and vary the number of trees $M$ to get different first-layer models. If we check the effective depth in the second-layer tree as shown in Figure S8, S9 and S10, we can also observe that the second layer tree will always choose the new feature to split. This verifies that Proposition 1 also holds in real world data sets.

**Convergence rate w.r.t. $M$.**    Figure S11 shows that DF enjoys a faster improvement in RMSE with the increasing of $M$ in these three real-world data sets. These experimental results match our theoretical analysis in Theorem 4 that DF has a faster convergence rate *w.r.t.* the number of trees $M$.

---

[1]http://lib.stat.cmu.edu/datasets/boston

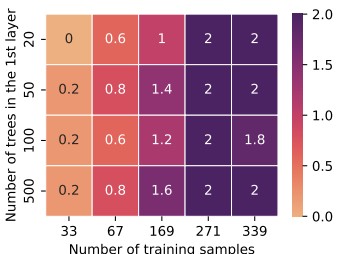

(a) Heatmap of the effective depth of new feature under different settings of the first layer forest.

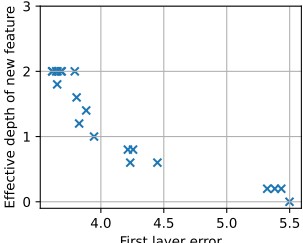

(b) Effective depth of new feature against the predictive error of the first layer.

Figure S8: Priority of new features in housing data set.

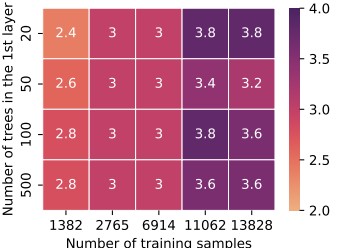

(a) Heatmap of the effective depth of new feature under different settings of the first layer forest.

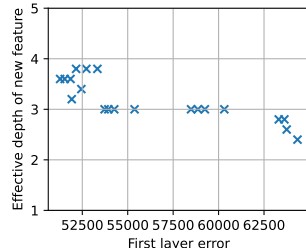

(b) Effective depth of new feature against the predictive error of the first layer.

Figure S9: Priority of new features in cadata data sets.

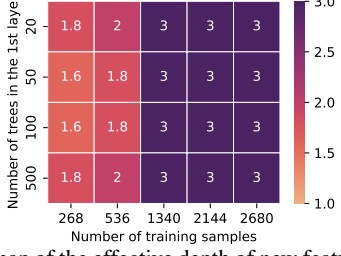

(a) Heatmap of the effective depth of new feature under different settings of the first layer forest.

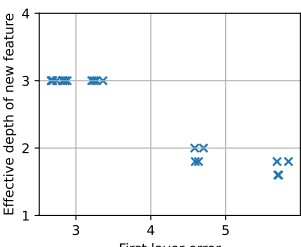

(b) Effective depth of new feature against the predictive error of the first layer.

Figure S10: Priority of new features in acoustic data set.

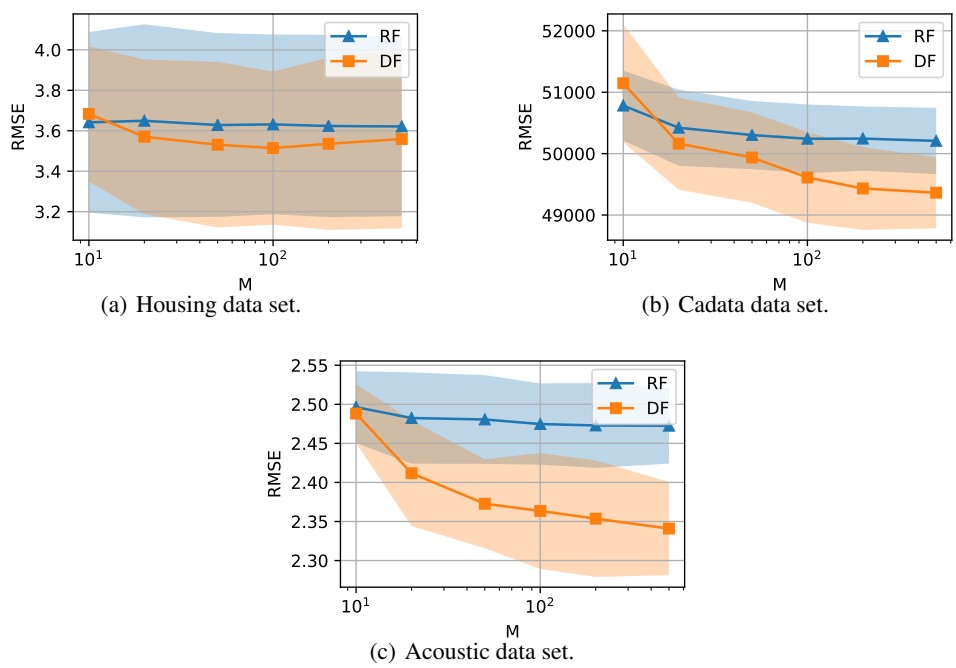

Figure S11: Root mean square error with the increasing of number of trees $M$ in real-world data sets.

# E    Dependence on label information in Assumption 2

In this section, we design a simple comparative experiment to show that the new features make the second-layer forest estimator much less dependent on label information. Specifically, we use a Completely Random Forest (CRF) to replace the random forest in the second layer of the deep forest, and CRF-split criterion does not depend on label information at all. The synthetic and real-world data sets used here are the same as Section 7.1, C and D. In Figure S12, we can find that the performance of the second-layer CRF can be close to the second-layer RF, and significantly outperforms the CRF trained on the original features. This implies that the new features play a positive role in reducing the model's dependence on label information.

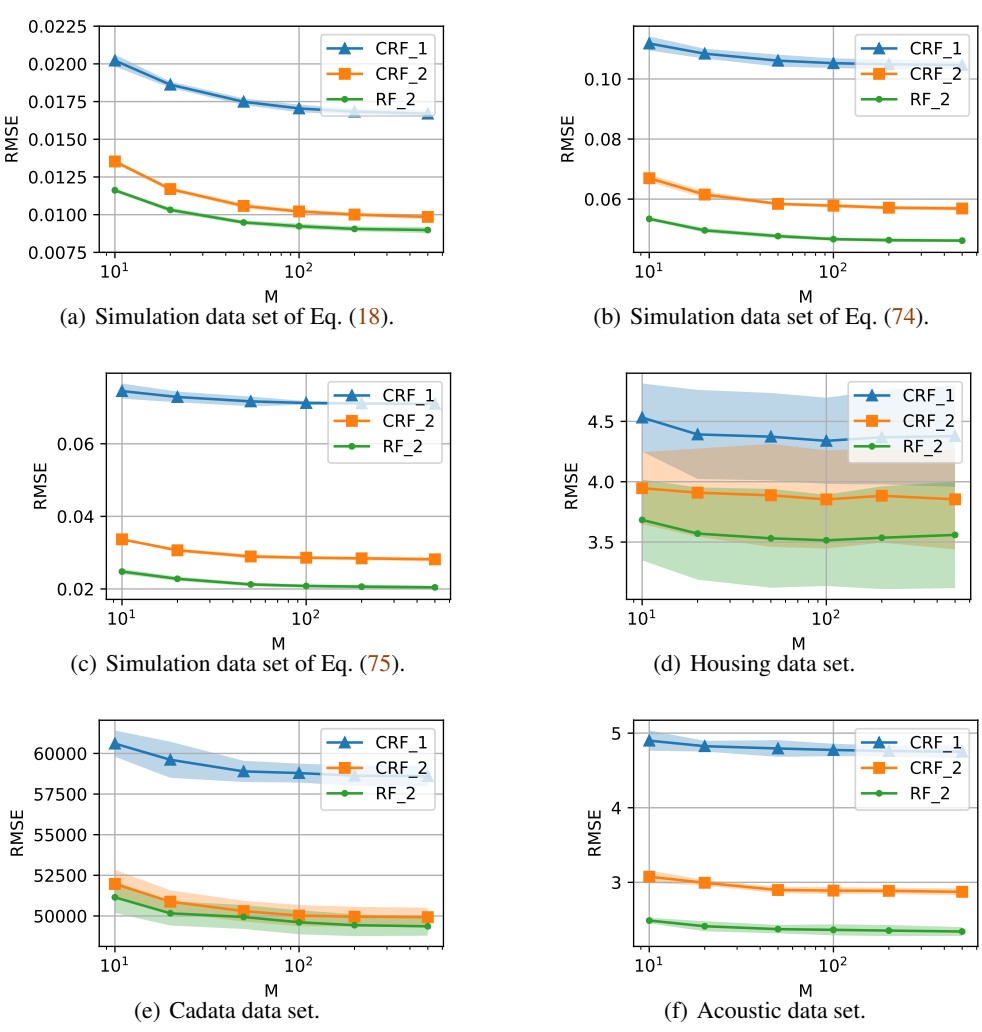

Figure S12: Root mean square error with the increasing of number of trees $M$. CRF_1 represents the completely random forest trained on the original feature space. CRF_2 represents the completely random forest trained on the new feature space. RF_2 represents the Breiman's random forest trained on the new feature space. As the number of samples in the dataset is larger, the performance of CRF and RF at layer 2 is more similar.