# OpenReview forum: "Depth is More Powerful than Width with Prediction Concatenation in Deep Forest"
_NeurIPS.cc/2022/Conference — NeurIPS 2022 Accept_

### Official Review · Reviewer_iuCL · 2022-07-02

**Rating:** 7
**Confidence:** 3
**Soundness:** 3 good
**Presentation:** 3 good
**Contribution:** 3 good

**Summary:**

The authors propose a theoretical analysis of deep random forests. Deep random forests stack layers of Breinman's random forest combined to completely random forest. The output of each layer in the regression setting is the average label of all sample falling in each leaf. Each layer takes as input a concatenation of the input features as well as previous features output by previous layers.

The authors focus on the two-layer deep forest and show multiple theoretical results aiming at explaining empirical performances.

Proposition 1 shows that stacking multiple layers make sense as the latter focus on the new features thus improving the regression function. Proposition 1 is proved in the infinite and finite regime (in the sample size).

Proposition 2 focuses on the variation of the true regression function f. The authors show that the variation of the regression function within a single cell is bounded in probability for any threshold and probability given that the sample size is large enough.

Theorem 3 recall the consistency of traditional random forests and states that the two layer deep forest are consistent as well. (given that the sample size per tree grows faster than the number of leaves which as stated by the authors is not what is usually done in practice).

Theorem 4 shows convergence speed to the infinite forest for the classical Breiman's forest and the two layer random forest. The former has a convergence rate of O(1/M) while the latter has O(1/M^2).

Concerning assumptions 1 and 2. Assumption 1 is a not too restrictive assumption probably used for simplicity in the proofs. Assumption 2 is less obvious to get at first sight and aims at showing that cells become less and less influenced by couple of samples labels and noise while the sample size tends to infinity.

Experiments on simulated data tend to confirm the theoretical claims and theorems of the paper.

**Questions:**

Would it be possible to give more elements about all the assumptions. Mainly:
- Do they apply in real life problem,
- What would we need to maintain the theoretical results while relaxing the assumptions ?

Would it be possible to evaluate experimentally the limit of these asumptions?

Equations 14 and 15 seem to have parenthesis problems.

**Limitations:**

Limitations of the work are clear from the core of the article but not particularly addressed in a specific area of the paper.

**Strengths And Weaknesses:**

- Strength
The paper is easy to read even for non-experts. Objects are well defined,
The paper proposes a theoretical justification for known empirical claims.
Deep forest is a recent contribution for which theoretical evidences lack and this paper contributes to the literature.

- Weaknesses
The assumptions, required for the proof, lack some analysis about their application in real life problems as well as motivation for their use.

---

> ### Author Response · Authors · 2022-08-02
> **Response to Reviewer iuCL**
>
> Thank you for your positive review.
>
> **Comment 3.1**: "Concerning assumptions 1 and 2. Assumption 1 is a not too restrictive assumption probably used for simplicity in the proofs. Assumption 2 is less obvious to get at first sight and aims at showing that cells become less and less influenced by a couple of sample labels and noise while the sample size tends to infinity. The assumptions, required for the proof, lack some analysis about their application in real-life problems as well as motivation for their use. Would it be possible to give more elements about all the assumptions? Mainly:
>
> - Do they apply to real-life problems?
>
> - What would we need to maintain the theoretical results while relaxing the assumptions?"
>
>
> **Response 3.1**: Thanks for your constructive suggestion. We will answer your concern about the two assumptions in this paper point by point.
>
> - Assumption 1 is a parametric model that is often used in applied statistics. We can only say that through a large number of statistical applications, we believe that there are many real-world problems suitable for modeling using Assumption 1, such as financial risk prediction, drug effect evaluation, etc. However, more general real-world problems (such as image classification, etc.) are difficult to strictly use assumption 1 to describe, but this is not the concern of this paper. Assumption 2 is to characterize the property of the forest model, and there are not too many restrictions on the data distribution of the real-world problem.
>
> - Under the theoretical framework of this paper, Assumption 1 is difficult to relax because its functional property is necessary. If Assumption 2 is relaxed, we need to further quantitatively analyze the trade-off between label information and partition randomness. And as far as we know, this is a very difficult problem, and there has been very little progress on this in the last two decades.
>
>
> ---
>
> **Comment 3.2**: "Would it be possible to evaluate experimentally the limit of these assumptions?"
>
> **Response 3.2**: Unfortunately, the statistics presented in Assumption 2 cannot be calculated experimentally. Further analysis of the trade-off between label information and partition randomness is a very interesting and challenging problem, and we will study it in future work. However, we have added an experiment to test the dependence of the second layer of random forest on label information in the revised manuscript **[Supplementary Section E, Page 27-28]**. The results show that when we replace RF with CRF (Completely Random Forest) whose split criterion is unsupervised, the performance of the second layer is still good. This shows that the new features reduce the influence of label information on the forest model, and also implies the rationality of Assumption 2 in deep forests.
>
> ---
>
> **Comment 3.3**: "Limitations of the work are clear from the core of the article but not particularly addressed in a specific area of the paper."
>
> **Response 3.3**: Thank you for your suggestion. We have added an explanation for the limitations of the paper in the revised manuscript. The details are as follows:
>
> > **[Supplementary Section F, Page 29, Line 756-763]** On the one hand, this paper focuses on the asymptotic consistency of deep forests, so the result is strictly true only when the number of samples tends to infinity. As the generalization analysis of deep forests with finite samples, we leave it to future work. On the other hand, The two assumptions used in this paper have certain limitations. Experiments on simulation and real-world data sets show that our theoretical results are valid in many objective function classes other than Assumption 1. How to further relax the conditions in Assumption 1 will be an interesting problem. As for Assumption 2, it is still not strictly verified. However, quantifying the trade-off between label information and partition randomness will be a very important topic in future work.
>
> *NOTE: Due to the limitation of the main content, this discussion section can only be temporarily placed in the supplementary. If this paper is accepted, we promise to include it in the main content.*
>
> ---
>
> ### Minor comments
>
> **Comment 3.4**: "Equations (14) and (15) seem to have parenthesis problems."
>
> **Response 3.4**: Thank you for pointing out this problem. It has been fixed in the revised manuscript.
>
> ---

---

### Official Review · Reviewer_QMfU · 2022-07-10

**Rating:** 5
**Confidence:** 4
**Soundness:** 3 good
**Presentation:** 3 good
**Contribution:** 1 poor

**Summary:**

In this paper, authors analyze the relationship between the depth and the number of trees of the deep forest. In recent years, deep forest has been widely studied due to its usability that combines the deep neural network with random forests. The authors validate their analysis through their experiments of the deep forests.

**Questions:**

Q1) Can you explain further why the analysis of the depth and the number of trees of the deep forest has a large impact? A more thorough explanation of why this analysis of deep forest is important should be added.

Q2) Does the experiment shown in this paper apply well to real vision or NLP applications? It would be nice to show the experimental results on more widely used task and dataset, but if that is not the case, authors need to provide evidence that the analysis in this paper can be applied to other applications.

**Limitations:**

There is no explanation for the limitations of the paper. Appropriate limitations should be included.

**Strengths And Weaknesses:**

Strength
+) Motivation and background studies are sufficiently provided.
+) Appropriate assumptions and mathematical derivations for analyzing deep forests are included.

Weaknesses
-) Analyzing the depth and the number of trees of the deep forests is limited to claim the novelty.
-) There is a lack of experiments with popular tasks and datasets, such as in computer vision or NLP domains.

----- after rebuttal -----

In rebuttal phase, authors revise their papers according to the comments. I understand the importance of the deep forest and the author's finding that its depth is more important than its width. However, I'm not sure if this finding would apply to real problems. In the revised paper, authors introduce new results on several real-world datasets and compared it to a random forest. However, it is necessary to show how much the existing SOTA results in the housing dataset can be increased by the authors' finding. For example, if this finding is applied to previous deep forest based methods, there should be  performance improvement compared to the baseline deep forests.

Considering the author's response, I increase the score to borderline accept by understanding that the authors have responded to reviewer's comments sufficiently, and that their findings are important.

---

> ### Author Response · Authors · 2022-08-02
> **Response to Reviewer QMfU (Part 2/2)**
>
> **Comment 2.2**: "There is a lack of experiments with popular tasks and datasets,... apply well to real vision or NLP applications? It would be nice to show the experimental results on a more widely used task and dataset, ... can be applied to other applications."
>
> **Response 2.2**: Thank you for your suggestion. In response to your concerns about the experimental results, we added some real-world data sets to confirm the theoretical results in the revised manuscript **[Supplementary Section D and E, Page 25-28]**. But we have to make the following statements:
>
> - The main contribution of this paper is the analysis of the consistency of deep forests. It means that when the number of samples tends to be infinite, the estimation function of the forest algorithm is equal to the target function, which is an **asymptotic property**. In real-world tasks, because the data distribution is very complex and the sample size is limited, the experimental results will not be as perfect as the simulation experiments. The **non-asymptotic generalization property** of random forests under finite samples is still an open problem, and we are willing to analyze it in future work.
>
> - CV and NLP are not scenarios where tree-based models are suitable for the application. Besides CV and NLP, there are still many important real-world problems that need to be solved by machine learning algorithms [1,2]. Deep forest is proposed to improve the feature representation ability of tree-based models on **tabular data**. In this paper, **our aim is not to benchmark DF performances but to investigate instead their underlying mechanisms**, so we suggest that you can refer to [3,4] to find out more about the performance of deep forests on real-world tasks with finite samples.
>
>
> [1] Sumanta Basu, Karl Kumbier, James B. Brown, and Bin Yu. Iterative random forests to discover predictive and stable high-order interactions. Proceedings of the National Academy of Sciences, 115(8):1943–1948, 2018
>
> [2] Tianqi Chen and Carlos Guestrin. XGboost: A scalable tree boosting system. In Proceedings of the 22nd ACM SIGKDD International Conference on Knowledge Discovery and Data Mining, pages 785–794, 2016.
>
> [3] Zhi-Hua Zhou and Ji Feng. Deep forest. National Science Review, 6(1):74–86, 2019.
>
> [4] Yi-He Chen, Shen-Huan Lyu, and Yuan Jiang. Improving deep forest by exploiting high-order interactions. In IEEE International Conference on Data Mining, pages 1030-1035, 2021.
>
> ---
>
> **Comment 2.3**: "There is no explanation for the limitations of the paper. Appropriate limitations should be included."
>
> **Response 2.3**: Thank you for your suggestion. We have added an explanation for the limitations of the paper in the revised manuscript. The details are as follows:
>
> > **[Supplementary Section F, Page 29, Line 756-763]** On the one hand, this paper focuses on the asymptotic consistency of deep forests, so the result is strictly true only when the number of samples tends to infinity. As the generalization analysis of deep forests with finite samples, we leave it to future work. On the other hand, The two assumptions used in this paper have certain limitations. Experiments on simulation and real-world data sets show that our theoretical results are valid in many objective function classes other than Assumption 1. How to further relax the conditions in Assumption 1 will be an interesting problem. As for Assumption 2, it is still not strictly verified. However, quantifying the trade-off between label information and partition randomness will be a very important topic in future work.
>
> *NOTE: Due to the limitation of the main content, this discussion section can only be temporarily placed in the supplementary. If this paper is accepted, we promise to include it in the main content.*

---

> ### Author Response · Authors · 2022-08-02
> **Response to Reviewer QMfU (Part 1/2)**
>
> Thank you for your review.
>
> **Comment 2.1**: "Analyzing the depth and the number of trees of the deep forests is limited to claim the novelty. Can you explain further why the analysis of the depth and the number of trees in the deep forest has a large impact? A more thorough explanation of why this analysis of deep forest is important should be added."
>
> **Response 2.1**: Thank you for your critical assessment of our work. We have added a thorough explanation of the importance of our analysis in the revised manuscript. The details are as follows:
>
> > **[Supplementary Section F, Page 29, Line 733-746]** Tree-based ensemble algorithms have achieved remarkable success early on, and there is substantial theoretical work analyzing the effect of the number of trees on their generalization performance [1-6]. However, while deep forests further improve generalization performance, there is no theory to prove **the advantages brought by depth**. Therefore, studying the influence of depth is the theoretical cornerstone for **distinguishing deep forests from traditional random forests**. The results of this paper prove that in terms of consistency, deep forests are no longer traditional random forests, and depth significantly improves the consistency convergence rate w.r.t the number of trees.
> >
> > For example, in the well-known deep neural networks (DNNs), there are a lot of theoretical works to study the effect of depth and width on its representation ability and generalization performance, which show the theoretical **advantages of deep neural networks over shallow neural networks** [7-11]. These works all contribute to the understanding of deep learning and provide theoretical insight for designing algorithms.
> >
> > **REF:**
> >
> > [1] Leo Breiman. Random forests. Machine learning, 45(1):5–32, 2001.
> >
> > [2] Gérard Biau. Analysis of a random forests model. The Journal of Machine Learning Research, 13(1): 1063–1095, 2012.
> >
> > [3] Misha Denil, David Matheson, and Nando De Freitas. Narrowing the gap: Random forests in theory and practice. In Proceedings of the 30th International conference on machine learning, pages 665–673, 2014.
> >
> > [4] Erwan Scornet, Gérard Biau, and Jean-Philippe Vert. Consistency of random forests. The Annals of Statistics, 43(4):1716–1741, 2015.
> >
> > [5] Erwan Scornet. On the asymptotics of random forests. Journal of Multivariate Analysis, 146:72–83, 2016.
> >
> > [6] Wei Gao and Zhi-Hua Zhou. Towards convergence rate analysis of random forests for classification. In Advances in Neural Information Processing Systems 33, pages 9300–9311, 2020.
> >
> > [7] Ronen Eldan and Ohad Shamir. The power of depth for feedforward neural networks. In Conference on learning theory, pages 907–940, 2016.
> >
> > [8] Matus Telgarsky. Benefits of depth in neural networks. In Conference on Learning Theory, pages 1517-1539, 2016.
> >
> > [9] Itay Safran and Ohad Shamir. Depth-width tradeoffs in approximating natural functions with neural networks. In International Conference on Machine Learning, pages 2979–2987. PMLR, 2017.
> >
> > [10] Eran Malach and Shai Shalev-Shwartz. Is deeper better only when shallow is good? In Advances in Neural Information Processing Systems, pages 6429–6438, 2019a.
> >
> > [11] Amit Daniely and Eran Malach. Learning parities with neural networks. In Advances in Neural Information Processing Systems, pages 20356-20365, 2020.
>
> *NOTE: Due to the limitation of the main content, this discussion section can only be temporarily placed in the supplementary. If this paper is accepted, we promise to include it in the main content.*
>
> ---

---

> ### Comment · Area_Chair_au8t · 2022-08-06
> **Please response to the author feedback**
>
> Dear Reviewer,
>
> Thanks for your efforts in reviewing the paper. Right now what I can see from your comments is that the paper doesn’t fit well your taste, but the technical novelty, soundness and potential impact of the paper are not criticized. Could you please look at the author’s rebuttal as well as the other reviews to see if you want to modify your reviews?
>
> Best,
> AC

---

> ### Author Response · Authors · 2022-08-07
> **Response to Reviewer QMfU (after rebuttal)**
>
> --- after rebuttal ---
>
> Dear reviewer QMfU,
>
> Thank you for reading our response and increasing the score. We noticed that you have raised some new questions, and we hope the responses below will resolve your concerns about our work.
>
> ---
>
> **Comment 2.4:** "...depth is more important than its width. However, I'm not sure if this finding would apply to real problems."
>
> **Response 2.4**: **[Supplementary Section D, Page 25-27]** The newly added experiment confirms that the theoretical findings of this work can be applied to real problems.
>
> ---
>
> **Comment 2.5:** "However, it is necessary to show how much the existing SOTA results in the housing dataset can be increased by the authors' finding. For example, if this finding is applied to previous deep forest based methods, there should be performance improvement compared to the baseline deep forests."
>
> **Response 2.5**: What this work analyzes is the depth advantage of the baseline deep forest compared to random forest, so this work does not propose a new deep forest algorithm. The purpose of this work is to analyze deep forests' theoretical properties, not to improve their performance. We will further explore how to improve the performance of deep forests in future work, which will be an important direction.
>
> ---
>
> Thanks again for your insightful suggestions on our work!

---

### Official Review · Reviewer_8wUS · 2022-07-11

**Rating:** 8
**Confidence:** 3
**Soundness:** 4 excellent
**Presentation:** 4 excellent
**Contribution:** 3 good

**Summary:**

The paper "Depth is More Powerful than Width in Deep Forest" studies the behavior of Deep Forest and shows that Deep Forests are consistent with a rate of $\mathcal O(1/M^2)$ where $M$ is the number of trees in the forest. To do so, the paper studies two-layered DF and its convergence in two different scenarios, one in which the number of data points is much larger than the number of leaves and  second when they are comparable equal. The resulting two Theorems establish the consistency of DF. Last, another Theorem establish the convergence speed of DF in terms of the number of trees which can be interpreted as "depth is more powerful than width". Experiments on synthetic data further supports the claim

**Questions:**

- It seems that I can apply Theorem 2-4 recursively if I have a Deep Forest with more than layers. Is this correct, and if so, does this improve the convergence rate to $\mathcal O(1/M^3), \mathcal O(1/M^4), \dots$ with each layer? This seems like an unusual strong result. Is this correct?

**Limitations:**

I think the limitations are addressed adequately.

**Strengths And Weaknesses:**

Overall I enjoyed reading the paper and I cannot criticize much about it. It is generally well-written and discusses an important topic. Deep Forest is arguably one of the more modern ensemble approaches (Ensembles of Neural Networks aside). While some theory has been developed for DF, it is not fully explored yet and hence the topic is of great interest. The paper heavily relies on results established for Random Forests, but extends them in a meaningful and non-trivial way. While the results are consistent with what one expect intuitively, I think it is important that they are tackled by rigors theory. Surprisingly, the results in the paper seem to be quite close to the practical application of DF. In summary:
- (+) Extends known results about Random Forest to Deep Forest thereby explaining some of the success of DF
- (+) Comparably easy to follow for such a theoretical paper. I was especially happy about the remarks and proof sketches, although I admit that I did not follow all proofs in the appendix.
- (-) I think a discussion on DF with more than two layers would have been interesting, although it seems straight-forward to apply Theorem  2 and 3 recursively (see question below)

---

> ### Author Response · Authors · 2022-08-02
> **Response to Reviewer 8wUS**
>
> Thank you for your positive review.
>
> **Comment 1.1**: "I think a discussion on DF with more than two layers would have been interesting, although it seems straightforward to apply Theorem 2 and 3 recursively. It seems that I can apply Theorem 2-4 recursively if I have a Deep Forest with more than two layers. Is this correct, and if so, does this improve the convergence rate to $\mathcal{O}(1/M^3)$, $\mathcal{O}(1/M^4)$ with each layer? This seems like an unusually strong result. Is this correct?"
>
> **Response 1.1**: Thank you for your insightful assessment. We add a discussion of this interesting result to the revised manuscript. The details are as follows:
>
> > **[Supplementary Section F, Page 29, Line 747-755]** In the infinite sample regime ($n=\infty$), the theoretical CART-split criterion can ensure that the residual error of each layer of random forest estimation is reduced at a rate of $1/M$, so the convergence rate of the total deep forest can be improved to $\mathcal{O}(1/M^3)$, $\mathcal{O}(1/M^4)$ with each layer. When we consider the consistency property, i.e., $n\to \infty$, the establishment of Theorem 4 depends on the existence of $N\in\mathbb{N}$ in Proposition 2. The value of $N$ varies with the layer, and may even grow exponentially (the specific rate is difficult to analyze). For example, in the second layer, we need $n>N=20000$ to make the result of Theorem 4 be simulated well, then in the third layer we may need $n>N=20000^2$ to make it well. Therefore, this result can only be established under the ideal condition ($n=\infty$) and it is difficult to be verified empirically.
>
> *NOTE: Due to the limitation of the main content, this discussion section can only be temporarily placed in the supplementary. If this paper is accepted, we promise to include it in the main content.*
>
> ---

---

> > ### Comment · Reviewer_8wUS · 2022-08-08
> > **Response to Response to Reviewer 8wUS**
> >
> > Thank you for answering my question. I think this part is a valuable addition to the paper and makes it somewhat easier to understand.

---

### Author Response · Authors · 2022-08-02
**General response**

**Response to all the reviewers**：We thank all the reviewers for their constructive assessment of our work. In the following, we address their concerns point by point.

**NOTE:** It seems that OpenReview MathJax viewer has some trouble parsing long formulas. If this happens on your browser, we recommend that you use other tools that can parse them online.

---

### Meta-Review · Area_Chair_au8t · 2022-08-20

**Recommendation:** Accept
**Confidence:** Certain

**Metareview:**

This paper establishes a set of theoretical results for characterizing the behavior of Deep Forest, an important model in deep learning. The analysis approach is novel and the results are of significant importance. The majority of the reviewers appreciate the authors’ contribution and all reviewers recommend acceptance. Thus I would strongly recommend accepting the paper.

**Award:**

No

---

### Decision · Program_Chairs · 2022-09-14

Accept